# Quantification of ligand and mutation-induced bias in EGFR phosphorylation in direct response to ligand binding

Daniel Wirth [1], Ece Özdemir[1] & Kalina Hristova [1] ✉

Signaling bias is the ability of a receptor to differentially activate downstream signaling pathways in response to different ligands. Bias investigations have been hindered by inconsistent results in different cellular contexts. Here we introduce a methodology to identify and quantify bias in signal transduction across the plasma membrane without contributions from feedback loops and system bias. We apply the methodology to quantify phosphorylation efficiencies and determine absolute bias coefficients. We show that the signaling of epidermal growth factor receptor (EGFR) to EGF and TGFα is biased towards Y1068 and against Y1173 phosphorylation, but has no bias for epiregulin. We further show that the L834R mutation found in non-small-cell lung cancer induces signaling bias as it switches the preferences to Y1173 phosphorylation. The knowledge gained here challenges the current understanding of EGFR signaling in health and disease and opens avenues for the exploration of biased inhibitors as anti-cancer therapies.

Receptor tyrosine kinases (RTKs) are single pass membrane receptors which control cell growth, differentiation, motility, survival, and metabolism. They have been implicated in many diseases, and are valuable drug targets. RTKs transduce biochemical signals via lateral interactions in the plasma membrane, by forming catalytically active dimers[1–3]. RTK dimerization, which is modulated by ligand binding, brings the kinase domains together in close proximity so they cross-phosphorylate each other on tyrosines in the activation loop[3,4]. This activates the kinases and they phosphorylate additional tyrosines which serve as binding sites for effector molecules, thus triggering downstream signaling cascades. Recent work has suggested that the signaling pathways originating from different RTK tyrosines can be differentially activated by different ligands - a phenomenon known as ligand bias[4].

Ligand bias has been studied primarily in the context of G-protein coupled receptors[5–7], where it has been shown to originate in the first step of signal transduction across the plasma membrane, i.e., due to differential signal propagation across the length of the receptor[8–10]. For RTKs, the origin of ligand bias has been debated[11,12]. Furthermore, experiments meant to explore RTK ligand bias have thus far probed for functional selectivity[13,14], which is "the combined effect of ligand and

system bias"[15]. While ligand bias is universal and pertains to all cell types as it depends on receptors and ligands, its manifestation may be different in different cells and tissues due to the system bias[15]. System bias is determined by the cellular/tissue/ physiological state context, including the identities of downstream signaling effectors in cells[15]. For RTKs, it has been further shown that the abundances of signaling effectors can introduce system biases[16]. Importantly, system bias can be perceived even at the level of RTK phosphorylation, which can be affected by feedback loops that operate within the cell[15,17]. A recent review of the current state of the field[15] emphasizes that "biased signaling represents very complex pharmacology, making experiment design, interpretation and description challenging and often inconsistent−causing confusion about what has really been measured and what can be concluded".

Here we introduce a methodology to measure ligand bias in direct response to ligand binding, thus simplifying interpretation and gaining insights into the origin of the bias. We show that we can identify and quantify ligand bias without contributions from feedback loops or system bias; we call it intrinsic ligand bias. We also show that we can identify and quantify signaling bias that is induced by an RTK

---

[1]Department of Materials Science and Engineering and Institute for NanoBioTechnology, Johns Hopkins University, 3400 Charles Street, Baltimore, MD 21218, USA. ✉e-mail: kh@jhu.edu

pathogenic mutation. The described methodology utilizes automatic imaging and data processing, and can be used for high-throughput screening of biased inhibitors to eliminate the deleterious effects of pathogenic RTK mutations.

## Results

### A model system to measure RTK phosphorylation in direct response to ligand binding

To be able to measure RTK phosphorylation in direct response to ligands without contribution from feedback loops and system bias, we used plasma membrane derived vesicles produced via osmotic vesiculation[18,19]. Such vesicles are produced from cells that have been transfected with genes encoding RTKs labeled with fluorescent proteins, and are imaged in a confocal microscope. As described in Supplementary Methods, we developed a neural network approach that allows high-throughput vesicle analysis. Once vesicles are identified, their membrane intensity is quantified as shown in Supplementary Fig. 1. The fluorescence intensities inside and outside the vesicles are quantified as well.

While all plasma membrane derived vesicles are known to have defects that allow the passage of macromolecules through the membrane[20], the vesicles produced via osmotic vesiculation allow the passage of very large macromolecules[19]. As a result, cytoplasmic signaling proteins such as Grb2-GFP (MW = 60 kDa) and PLCγ-GFP (MW = 210 kDa) diffuse through the vesicle membranes and become infinitely diluted in the buffer which is contiguous with the vesicle lumens[19]. So do cytoplasmic proteins that bind to the lipids on the cytoplasmic side (such as PLCδ-PH-GFP), or to RTKs on the cytoplasmic side (such as PLCγ-GFP). These proteins dissociate from the membrane because their residence times on the membrane are much shorter than the timescale of vesicle production, ~12 h[19]. Thus, these vesicles lack the components of signaling feedback loops.

Previous work has shown that the lipid composition of the vesicles is very similar to the lipid composition of the plasma membrane[19]. Here we investigated the permeability of these vesicles to FITC-labeled dextrans (20–2000 kDa) which were added externally after vesicle production. The FITC intensity inside and outside the vesicles were measured after 1 h to quantify the degree of dextran penetration through the vesicle membrane. Figures 1A and B show that the 20 kDa and 70 kDa dextrans equilibrate across the vesicle membrane without any obstruction (intensity ratio: $1.00 \pm 0.03$ and $0.98 \pm 0.03$, respectively). This confirms the presence of large defects in the membrane, and explains the lack of retention of cytoplasmic signaling proteins. Reduced penetration of dextrans is observed starting from molecular weight 250 kDa (intensity ratio: $0.84 \pm 0.05$). Dextrans are known to form rod-like structures in aqueous solutions[21], and hydrodynamic radii for the 20, 73, 250, 500, and 2000 kDa dextrans have been reported as 3.2, 6.5, 11.5, 15.9, and 26.9 nm, respectively[22]. Taking into account that the hydrodynamic radius for an IgG antibody (150 kDa) is 5.4 nm, and thus falls in between the hydrodynamic radii of 20 kDa and 70 kDa dextran, we predict that antibodies which are added externally will penetrate the vesicles. This makes it possible to detect specific phosphotyrosines on the kinase domain of an RTK in the vesicle lumen as illustrated in Supplementary Fig. 4. The antibodies are recruited to the vesicle membrane upon tyrosine phosphorylation, where their fluorescence intensities are measured.

Vesicles were produced from cells transfected with EGFR. Experiments were set up with 100 nM EGF in the presence of an ATP cocktail containing $Mg^{2+}$ and a phosphatase inhibitor (see Methods). A FITC-labeled anti-pY 4G10 antibody, which recognizes any phosphorylated tyrosine residue on EGFR in response to EGF stimulation, was used for detection. An IgG-FITC isotype control antibody was used as a control. While no significant binding to the vesicle membrane was detected for the control antibody, a clear increase in

membrane fluorescence was observed for the anti-pY antibody upon EGF addition (Fig. 1C). In both cases the solution fluorescence intensity of the antibody fluorophores is equal inside and outside the vesicles, showing that antibodies are able to freely diffuse across the vesicle membrane.

The maximum antibody binding and therefore receptor phosphorylation is achieved after about 20 min, with a time course which is likely affected by ligand and antibody diffusion (Fig. 1D). Importantly for this work, a plateau in the fluorescence was observed, demonstrating that the phosphorylation comes to equilibrium, consistent with the expectation that no feedback loops are present. As a control, no increase in fluorescence was observed in the absence of ligand and ATP kinase cocktail.

### Quantification of intrinsic ligand bias in EGFR signal propagation across the plasma membrane

We investigated if there is preference for the phosphorylation of one of two tyrosines when EGFR is activated by three EGFR ligands: EGF, TGFα, and epiregulin. The two tyrosines that were probed, Y1068 and Y1173, are in the long unstructured tail of EGFR and have profound importance for signaling. Phosphorylation of Y1068 leads to the recruitment of Grb2 and Gab1 and the activation of AKT and STAT3/5 signaling pathways[23–25]. On the other hand, Y1173 phosphorylation leads to the recruitment of Shc and the activation of the MAPK/ERK signaling cascade (although there is cross-talk between the different pathways which is cell-specific)[23]. The differential phosphorylation of these two tyrosines is believed to lead to different functional outcomes, and their differential phosphorylation in cells has already been used as an indicator of functional selectivity in EGFR signaling[16,25].

Experiments were performed with EGFR-mTurquoise (EGFR-mTurq), in which the fluorescent protein mTurq was attached to the C-terminus of EGFR via a 15 aa linker. This attachment does not impact the activation of EGFR[26]. The cells were vesiculated and thousands of individual vesicles were imaged. To detect EGFR phosphorylation, we used either anti-pY1068 or anti-pY1173 EGFR antibody, labeled with AlexaF488. The molar concentration of the antibodies always exceeded at least 5 times the total molar concentration of EGFR (~10 nM) and the pY-antibody dissociation constant (low nM). To start the reaction, we added ligands together with ATP kinase cocktail (1 mM ATP, 0.5 mM DTT, 10 mM $MgCl_2$, 0.1 mM $Na_3VO_4$ (a phosphatase inhibitor)). The antibody was recruited to the vesicle membrane and the recruitment was quantified through the increase in membrane fluorescence. The imaging was performed at least 1 h after the beginning of the reaction, based on kinetic traces of single vesicles which show complete equilibration after ~20 min. Each vesicle was imaged using an automated microscope stage in two scans: one exciting mTurq at the C-terminus of EGFR, to assess EGFR concentration in each vesicle, and one exciting the fluorophore on the anti-pY antibody, to assess its concentration on the membrane in each vesicle. The degree of phosphorylation, per EGFR molecule, is thus proportional to the fluorescence ratio in the antibody channel and the EGFR channel.

Complete dose-response curves for WT EGFR Y1068 and Y1173 phosphorylation per EGFR molecule, in response to EGF, TGFα, and epiregulin, were collected (Fig. 2A). A total of 11,570 vesicles were imaged, while the concentration of different ligands was varied from zero to saturating concentrations. For each individual vesicle, on the y axis we report the ratio of: (i) the fluorescence of AlexaF488, linked to the anti-pY antibody and (ii) the fluorescence of mTurq, linked to the receptor. Thus, on the y axis the values are proportional to the degree of EGFR phosphorylation, and on the $x$ axis is the ligand concentration (Fig. 2A). As all measurements utilized the same microscope setting and the same antibody batches, all Y1068 data are on the same scale and all Y1173 data are on the same scale.

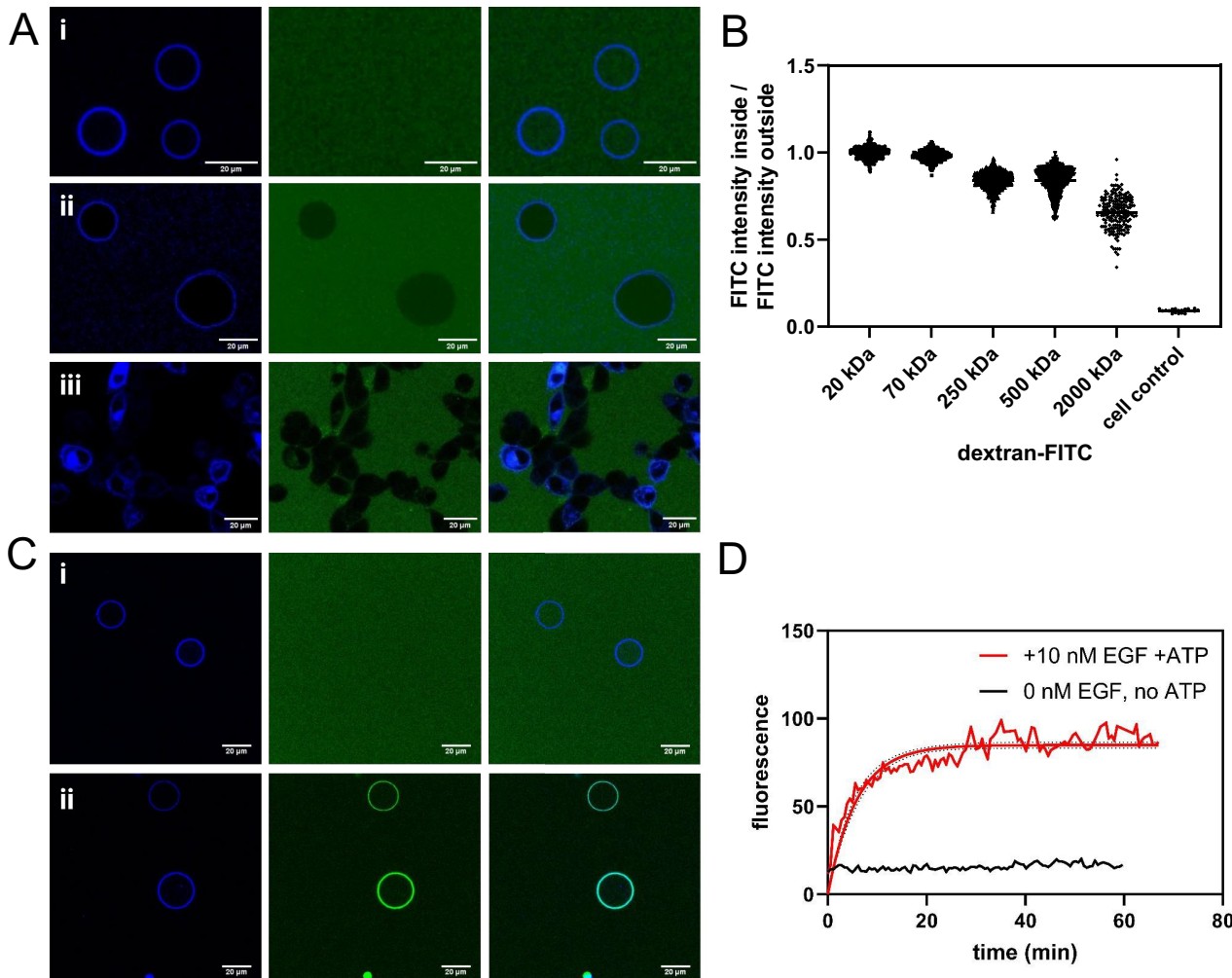

**Fig. 1 | Plasma membrane derived vesicles as a tool to probe RTK phosphorylation in response to ligand binding. A** Determination of the size cutoff for equilibration across the vesicle membrane. FITC-labeled dextrans of different molecular weights were added to vesicles with FGFR3-mTurquoise in their membrane. Left column: receptor, middle column: dextran, right column: overlay. (i) 70 kDa dextran. The intensity inside the vesicle and outside is the same. (ii) 2000 kDa dextran. The intensity inside the vesicle is lower. Shown are representative images from 11 independent vesicle experiments. (iii) cells, EGFR-mTurquoise + control antibody (rat IgG2bκ-FITC). The antibody does not cross the cell membrane. Shown are representative images from 3 independent cell experiments. **B** Intensity ratios between FITC-labeled dextran inside and outside of vesicles for dextrans of different molecular weights. Each data point represents the ratio for one vesicle. Data are for 3454 vesicles in 14 independent experiments. **C** EGFR phosphorylation in vesicles. First column: receptor channel, middle column: antibody channel, right column: overlay. (i) Vesicles derived from CHO cells with EGFR-mTurq incorporated into the vesicle membrane were incubated with 10 μg/mL IgG-FITC isotype control antibody. (ii) Vesicles derived from CHO cells with EGFR-mTurq incorporated into the vesicle membrane were incubated with 100 nM EGF, ATP/salt cocktail, and FITC anti-pY 4G10 antibody. The antibody fluorescence can be seen on the membrane. Representative images from 3 independent experiments with the 4G10 antibody. **D** Phosphorylation signal on the vesicle membrane over time. The fluorescence of the antibody was measured on the membrane of a single vesicle over time in response to 10 nM EGF with added ATP cocktail (red line). The black line shows a control experiment in the absence of EGF and ATP.

To determine if either Y1068 or Y1173 is preferentially phosphorylated by a ligand in comparison to another ligand, or whether there is no preference, we created bias plots (Fig. 2B). The bias plots depict three comparisons for the three ligands (TGFα vs EGF (reference), epiregulin vs EGF (reference), and epiregulin vs TGFα (reference)) and the two tyrosine phosphorylation responses. In the bias plots one response (pY1173) is plotted against the second response (pY1068) at the same ligand concentrations. These bias plots report directly on the relative effectiveness of the ligands to produce the two responses, without the need for assumptions or mathematical modeling[5,12]. The bias plots in Fig. 2B are different, indicating that there is bias. In particular, the epiregulin points diverge from the EGF and TGFα points, in the direction of Y1173 phosphorylation. Thus, the bias plot in Fig. 2B demonstrates that epiregulin induces preferential phosphorylation of Y1173 over Y1068 when compared to EGF and TGFα.

Signaling bias due to a specific ligand can be identified and quantified with respect to a reference ligand by also calculating bias coefficients[7,15,27,28], such as the widely used $\beta_{lig}$ given in Eq. (1)[7,12]. This requires that we know the potencies, $EC_{50}$, and the efficacies (maximum effects), $E_{top}$, of the ligand and the reference ligand for the two responses, pY1068 and pY1173. The coefficient $\beta_{lig}$ has a sign that indicates the preference of the ligand, as compared to the reference ligand, for a particular response (+, if the first response is preferred and −, if the second response is preferred), as well as a magnitude which reports on the degree of bias. The case of $\beta_{lig} = 0$ indicates that the ligand is not biased when compared to the reference ligand.

To calculate $EC_{50}$ and $E_{top}$ for the three ligands, we first note that the phosphorylation at zero ligand in Fig. 2A is not zero. This is consistent with prior work, showing that EGFR can be phosphorylated in

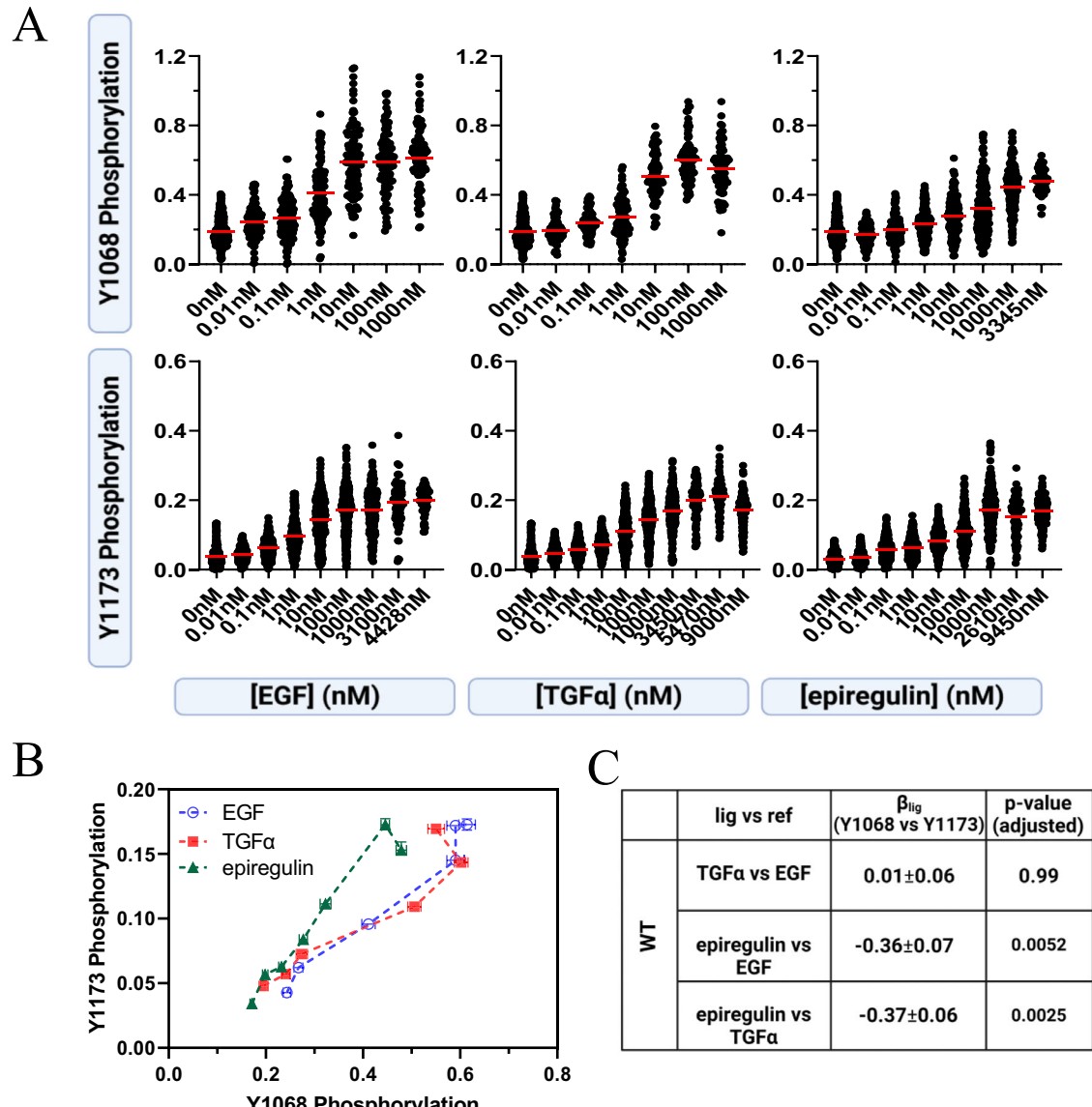

**Fig. 2 | Ligand bias for WT EGFR. A** Raw dose response curves for Y1068 and Y1173 phosphorylation per EGFR molecule, for the ligands EGF, TGFα, and epiregulin. Each point represents the ratio of either anti-pY1068 or anti-pY1173 fluorescence and EGFR-mTurq fluorescence for one individual vesicle. Each curve contains ~1000 to ~3000 data points (single vesicles). **B** Bias plots. Shown are means (symbols) and standard errors (often smaller than symbols). The epiregulin points diverge from the EGF and TGFα points. In total, data are from 11,570 single vesicles over 25 independent experiments. **C** Bias coefficients and standard errors. Epiregulin is biased toward Y1173 phosphorylation as compared to EGF and TGFα. Ordinary one-way ANOVA, followed by Tukey's test, was used to determine statistical significance. The $p$ values are adjusted for multiple comparisons.

the absence of ligand, but cannot trigger downstream signaling[26], likely because the structure of the unliganded EGFR dimer is different from the structure of the active ligand-bound EGFR dimers[26]. To characterize the response to ligand, we corrected the measured dose responses for the contribution of the unliganded dimers as described in the Supplement (Supplementary Fig. 7). The best-fit potencies (EC$_{50}$) and the efficacies (E$_{top}$), calculated using a Hill slope of 1, are given in Supplementary Table 1. The corrected averaged dose response curves are shown in Supplementary Fig. 8. The three bias coefficients in Fig. 2C are calculated for the three comparisons (TGFα vs EGF (reference), epiregulin vs EGF (reference), and epiregulin vs TGFα (reference)) using eqn 1. A one-way ANOVA analysis of the bias coefficients in Fig. 2D, followed by Tukey's multiple comparison test, demonstrated the presence of epiregulin-induced bias towards Y1173 phosphorylation when compared to EGF ($p = 0.005$) and TGFα ($p = 0.003$). This analysis is entirely consistent with the bias plots,

which are created by plotting the experimentally measured data without any corrections or assumptions.

## The common NSCLC L834R (L858R) driver mutation in EGFR induces intrinsic bias in signal propagation across the plasma membrane

NSCLC represents over 85% of all lung cancers and is associated with high mortality[29]. The 5-year survival for all stages of progression is <17%. This cancer is due to EGFR mutations in ~10–15% of Caucasian patients and in up to 50% of Asian patients. Of the single amino acid mutations, the L834R mutation is the most common one, accounting for about 40–45% of the cases where EGFR is mutated. (This mutation is often referred to as the "L858R mutation" when the EGFR signal peptide is counted).

We acquired dose response curves for L834R EGFR in response to EGF, TGFα, and epiregulin. A total of 8009 individual vesicles were imaged and analyzed in these experiments (Fig. 3A). We then

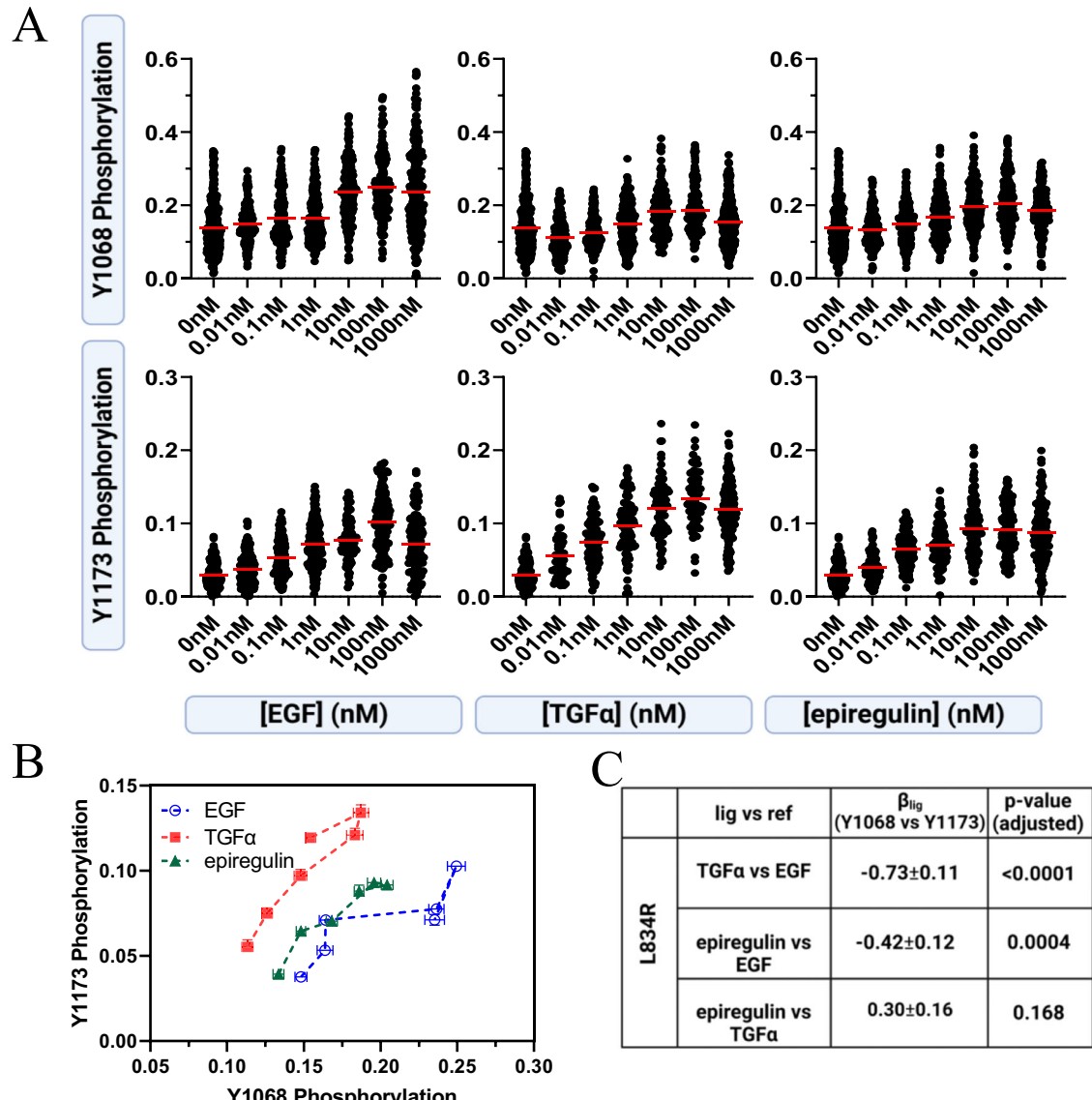

**Fig. 3 | Ligand bias in L834R EGFR phosphorylation. A** Single vesicle dose response curves for Y1068 and Y1173 phosphorylation per EGFR molecule, for EGF, TGFα, and epiregulin. Each point represents the ratio of either anti-pY1068 or anti-pY1173 fluorescence and EGFR-mTurq fluorescence for one individual vesicle. Each curve contains ~1000 to ~1800 data points. **B** Bias plots. Shown are means (symbols) and standard errors (often smaller than symbols). In total, data are from 8009 vesicles in 23 independent experiments. **C** Bias coefficients and their standard errors. Ordinary one-way ANOVA, followed by Tukey's test, was used to determine statistical significance. The *p* values are adjusted for multiple comparisons.

constructed ligand bias plots (Fig. 3B). In the case of L834R EGFR, both TGFα and epiregulin are biased toward Y1173 phosphorylation over Y1068 phosphorylation, as compared to EGF. Thus, the relative bias of the three EGFR ligands is altered due to the L834R mutation. These conclusion from the bias plots are supported by the calculations of bias coefficients (Fig. 3B) and their statistical analysis. The corrected averaged dose response curves are shown in Supplementary Fig. 8.

To directly answer the question if the mutation causes bias in EGFR signaling, we created bias plots while directly comparing the wild-type and the mutant (Fig. 4A–C). These are mutation-induced bias plots, distinctly different from the ligand bias plots in Fig. 2B and Fig. 3B, as they now directly compare the mutant and the wild-type. In Fig. 4A–C we see that the mutation induces significant preference for Y1173 phosphorylation over Y1068 phosphorylation, when compared to the wild-type, in the presence of the three ligands. We then calculated a different type of bias coefficient, the mutation-induced bias coefficient $\beta_{mut}$, to quantify the degree of bias introduced

by the mutation in EGFR signaling in response to a specific ligand. The values were calculated using Eq. (2), where "response A" refers to Y1068 phosphorylation and "response B" refers to Y1173 phosphorylation (Fig. 4D). The effect is largest in the case of TGFα, but highly statistically significant for all ligands, based on t-tests. This is a direct demonstration that the L834R mutation induces bias in EGFR phosphorylation in the plasma membrane for all studied ligands. The corrected averaged dose response curves for the wild-type and the mutant are compared in Supplementary Fig. 9.

**A measurement of the phosphorylation transducer function**
The transducer function relates a response to the stimulus that is causing it[30]. In our case, the response is the phosphorylation of a tyrosine in the intracellular domain of an RTK. The stimulus is the formation of the ligand-bound RTK dimers[30]. We therefore sought to measure both ligand binding and phosphorylation simultaneously so we can plot one vs the other and obtain the transducer function.

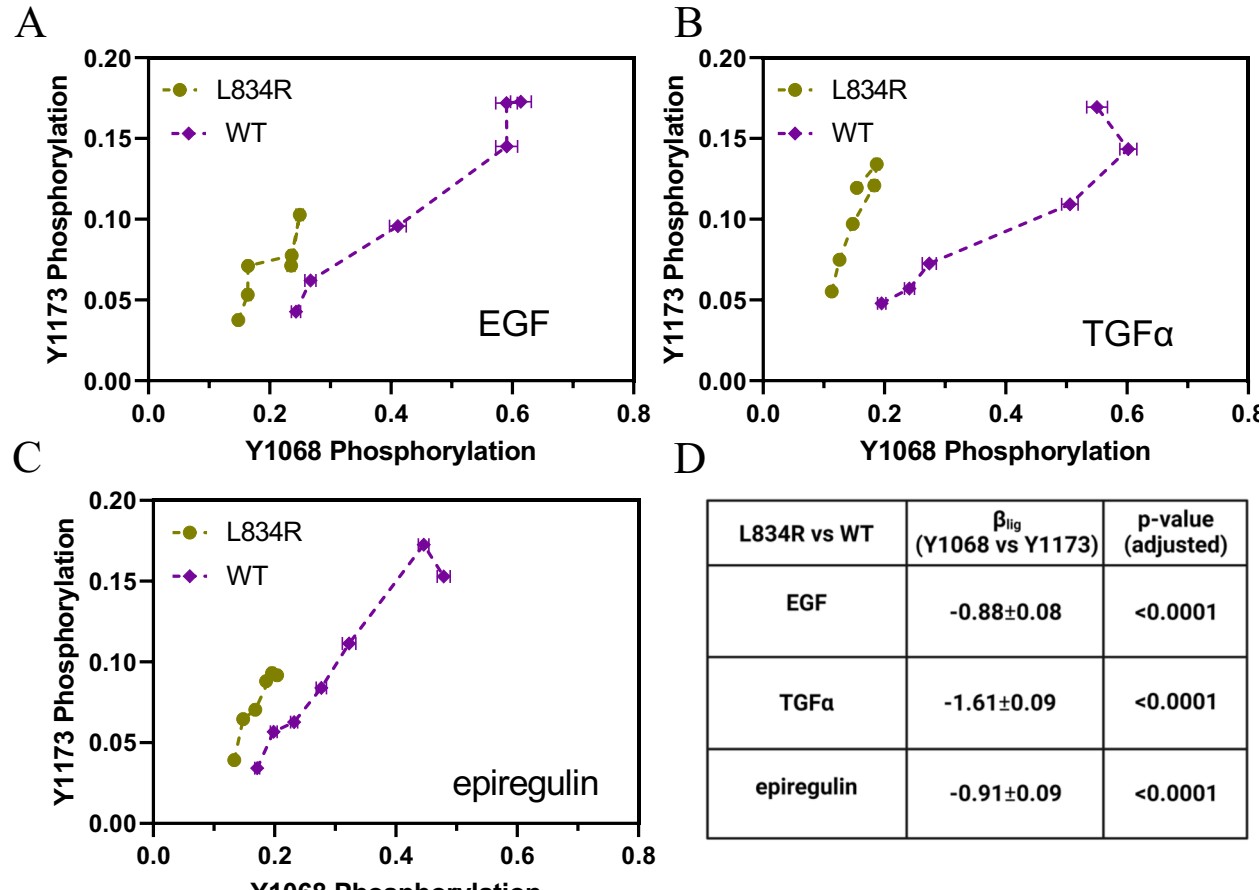

**Fig. 4 | L834R mutation-induced bias. A–C** L834R-induced bias plots in the presence of EGF, TGFα, and epiregulin. Shown are means (symbols) and standard errors (often smaller than symbols). In total, data are from 11,570 single vesicles over 25 independent experiments for WT and 8009 vesicles for L834R EGFR in 23 independent experiments. **D** Bias coefficients and their standard errors. Three pairwise comparisons were performed using two-tailed *t*-tests. The *p* values were adjusted for multiple comparisons using the Holm-Sidak correction. The L834R mutation induces statistically significant preference for Y1173 phosphorylation over Y1068 phosphorylation, as compared to the wild-type, in the presence of all three ligands.

To demonstrate the feasibility of transducer function measurements, we used commercially available EGF ligand from mouse that is labeled with rhodamine at its N-terminus (rho-mEGF, Thermofisher, E3481), a ligand that binds human EGFR with 3 times lower affinity than human EGF[31]. To determine the transducer function for rho-mEGF, individual vesicles were imaged in a confocal microscope in three scans to measure: (i) the fluorescence of rhodamine, linked to mEGF, on the membrane, to quantify the bound ligand in the plasma membrane in each vesicle Ex:552 nm; Em:565–625 nm, (ii) the fluorescence of AlexaF488, linked to the anti-phosphoY antibody, to quantify phosphorylated EGFR in the membrane Ex:488 nm; Em:500–540 nm (iii) the fluorescence of mTurq, linked to the receptor, in order to quantify EGFR in the plasma membrane in each vesicle Ex:448 nm; Em:460–510 nm. One vesicle, imaged in the three scans, in the presence of 5 nM EGF, is shown in Fig. 5A. More than 3000 vesicles were imaged, while the ligand concentration was varied from zero to saturating concentrations.

The mouse and human EGF differ in sequence and affinity to human EGFR (See Supplemental Methods). To assess if the mouse rho-EGF induces biased EGFR signaling, as compared to the three human ligands, we used the acquired phosphorylation dose response curves shown in Supplementary Fig. 10 to construct bias plots comparing rho-mEGF and the human ligands (Fig. 5B), and we calculated bias coefficients (Supplementary Table 3). By ANOVA, the two EGF ligands are not biased, despite the reported differences in affinity to EGFR[31].

In Fig. 5C, we plot the phosphorylation response (fluorescence in the antibody channel divided by the fluorescence in the EGFR channel) as a function of the stimulus (ligand-bound EGFR fraction, $f_{bound}$), for each individual vesicle that was imaged in the three channels. The x axis (fluorescence in the ligand channel divided by the fluorescence in the EGFR channel) is scaled such that the maximum average fraction of ligand-bound EGFR is set to 1. This plot represents the transducer function.

We fit the data in Fig. 5C using Eq. (5) to determine (i) Rmax, the maximal possible signal that can be achieved in the experiment by a true full agonist ($K_{resp} \rightarrow 0$), which depends on the fluorescent properties of the antibodies and (ii) $K_{resp}$, the fraction of ligand-bound receptors that yields 50% of $R_{max}$. The smaller the value of $K_{resp}$, the more efficient the phosphorylation. The best-fit values for Y1068 phosphorylation are $K_{resp} = 0.40 \pm 0.03$ and $R_{max} = 0.95 \pm 0.03$. The best-fit values for Y1173 phosphorylation are $K_{resp} = 0.86 \pm 0.08$ and $R_{max} = 0.39 \pm 0.02$. In Fig. 5D, we show the fits along with the data, binned in intervals of 0.1 on the x axis. Only bins containing at least 50 vesicles are shown.

$K_{resp}$ for Y1068 phosphorylation is the smaller of the two, indicating that Y1068 phosphorylation is more efficient than Y1173 phosphorylation in response to EGF. Since the two $R_{max}$ values differ because of the different fluorescent properties of the two antibodies, in Fig. 5E we plot the normalized transducer function, i.e., the dependence of $R_{phospho}/R_{max}$ on the bound fraction, $f_{bound}$. The y value at $f_{bound} = 1$ is the phosphorylation efficiency, calculated using Eq. (8) as

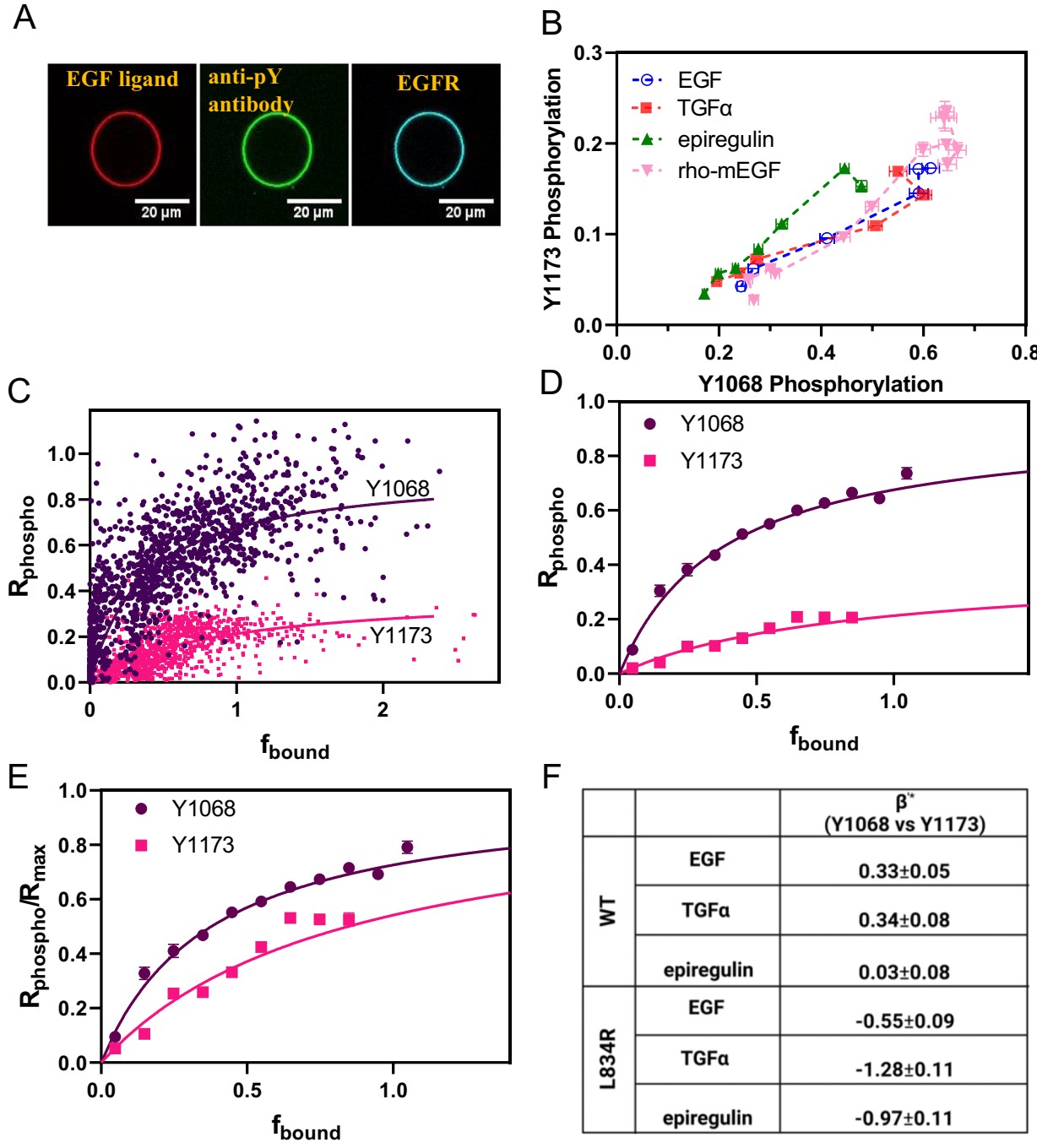

**Fig. 5 | The rho-mEGF EGFR transducer function. A** One vesicle imaged in three channels at 5 nM rho-mEGF in the presence of ATP/kinase cocktail. **B** Bias plots for rho-mEGFR and human EGF, TGFα, and epiregulin. EGF and rho-mEGF are not biased ligands. Shown are means and standard errors. In total, data are from 22,670 single vesicles over 34 independent experiments. **C** Phosphorylation response (fluorescence in the antibody channel divided by the fluorescence in the EGFR channel) vs ligand-bound EGFR fraction for individual vesicles (fluorescence in the ligand channel divided by the fluorescence in the EGFR channel). The *x* axis is scaled such that the maximum average bound fraction is set to 1, and the y axis is phosphorylation corrected for constitutive (ligand-independent) phosphorylation. The solid lines are the transducer function fits (Eq. (5)) to all the single vesicle data. Data are from 2173 individual vesicles for Y1068 and 1987 individual vesicles for Y1173 over 9 independent experiments in which the concentration of rho-mEGF was varied (**D**) The single-vesicle data has been binned in an interval of 0.1 and is shown along with the fits to all the single vesicle data. Shown are standard errors; if not visible they are smaller than the symbols. **E** The normalized transducer function, given by Eq. (7), and the calculated standard errors. Data are from 2173 individual vesicles for Y1068 and 1987 individual vesicles for Y1173 over 9 independent experiments. **F** Absolute bias coefficients calculated using Eqs.(19)–(23).

0.71 ± 0.02 for Y1068 phosphorylation and 0.54 ± 0.02 for Y1173 phosphorylation. Thus, rho-mEGF is a partial agonist for both responses. Human EGF, TGFα, and epiregulin are also partial agonists, as the $E_{top}$ values for their responses do not exceed the ones for rho-mEGF (see Supplementary Tables 1 and 2).

## Calculation of absolute bias coefficients

Bias coefficients calculated using Eqs. (1) and (2) are relative, i.e., $\beta_{lig}$ is always calculated with respect to the reference ligand in the literature. However, the effective equilibrium constant $K_{resp}$ can be used to calculate absolute bias coefficients, $\beta^{*}{}_{lig}$, and $\beta^{*}{}_{ref}$ (see Eqs. (22)

and (23). First, we calculate $\beta'^*_{rho-mEGF}$ using Eq. (20); $\beta'^*_{rho-mEGF} = 0.33 \pm 0.5$. Since this ligand is not biased in comparison to EGF, $\beta'^*_{EGF} = \beta'^*_{rho-mEGF}$. The absolute $\beta'^*_{EGF}$ directly reports on the preference of a ligand toward either Y1068 or Y1173 phosphorylation. The value of $\beta'^*_{EGF}$ is positive, indicating that Y1068 is preferentially phosphorylated in response to EGF, as compared to Y1173.

With the values of $\beta_{lig}$ and $\beta'^*_{EGF}$ known, we calculate the absolute bias coefficients $\beta'^*_{TGF\alpha}$ and $\beta'^*_{epiregulin}$, for TGFα and epiregulin, using Eq. (22). Similarly, we calculate the absolute bias coefficients $\beta'^*_{mut}$ for the L834R mutant using Eq. (23). All absolute bias coefficients are shown in Fig. 5F. We see that Y1068 in WT EGFR is preferentially phosphorylated in response to EGF and TGFα, but there is no preference in response to epiregulin. The signaling of the mutant is always biased toward Y1173 phosphorylation.

## Discussion

Here we introduce a methodology to quantify (i) ligand and mutation-induced bias coefficients, both on relative and absolute scales and (ii) the transducer function describing RTK phosphorylation upon ligand stimulation. These critical descriptors of RTK activation are measured in direct response of the RTKs to ligand binding, without contributions from downstream signaling feedback loops and system bias. This method can be used for all RTKs, and all membrane receptors in general.

The power of the methodology comes from the use of plasma derived vesicles produced via osmotic vesiculation. While these vesicles lack cytoskeleton and have perturbed asymmetry in their lipid composition, they allow access of macromolecules to both the extracellular and intracellular domains of the RTKs. In this respect, the plasma membrane derived vesicles can be considered as an alternative to nanodisks[32,33]. Unlike nanodisks, they do not impose artificial constraints on the free association of EGFR, and are a much more faithful mimic of the plasma membrane as they incorporate native lipids. They do not require RTK extraction out of the native plasma membrane and provide a contiguous membrane to ensure that the RTKs can associate with each other as they do in cells. Noteworthy, association constants measured for EGFR in vesicles and in cells are the same[26,34]. Also noteworthy, plasma membrane derived vesicles were recently leveraged in cryoEM studies to determine the high resolution structure of a membrane protein[35].

The use of vesicles allows us to make measurements of RTK phosphorylation in the absence of cytoplasmic molecules involved in downstream signaling and thus in the absence of system bias. The vesicles offer additional unique advantages. The phosphorylation reaction is initiated by the researcher, by adding ligand and ATP kinase cocktail, and phosphorylation is followed through the recruitment of labeled specific anti-pY antibodies to the vesicle membranes. There is no signal attenuation because there is no RTK downregulation. Soluble phosphatases are not present, and the membrane phosphatases are inhibited since the ATP kinase cocktail contains the inhibitors. Only mature RTKs in the plasma membrane are present. Antibodies, specific for only one tyrosine on only one RTK, verified in many RTK publications, are used in the detection. Data points in dose-response curves are derived from individual vesicles. Imaging is automated through the use of a commercial automated stage. Data processing is also automated using a neural network. The high-throughput format allows us to measure thousands of data points per dose response curve, and thus minimize random errors which arise due to white noise in imaging[36]. Thus, the experimental platform is suitable for high-throughput screening of RTK inhibitors.

The data acquired with this method can be compared to published data. First, epiregulin is known to have lower potency for WT EGFR phosphorylation, as compared to EGF and TGFα[11,34,37–39], in accordance with our measurements. Second, it is known that EGF and epiregulin signal differently through EGFR, since epiregulin induces cell differentiation under the same conditions where EGF induces proliferation[11]. Ligand bias leads to fundamentally different biological outcomes[5,15],

consistent with these prior findings and our observations of differential Y1068 and Y1173 phosphorylation. Thus, our results are consistent with knowledge in the literature. As an important development, we now construct bias plots, considered the most reliable proof of bias in the literature[5,15], and we calculate bias coefficients which support the bias plots. Thus, the degree of bias for multiple ligands is now quantified in the absence of feedback loops and system bias.

Another important result is the calculation of the EGF phosphorylation efficiency, which is the maximum possible phosphorylation that can be achieved in response to EGF. It is about 70% for Y1068 phosphorylation and 55% for Y1173 phosphorylation. This measurement is possible because ligand binding and EGFR phosphorylation are measured simultaneously, for hundreds of individual vesicles. The fit of the transducer function yields not only $K_{resp}$, used to calculate absolute bias coefficients, but also $R_{max}$, the maximum possible response to a ligand. We thus demonstrate that EGF is not a full agonist, which suggests that new ligands can be designed to more strongly activate EGFR.

We also gain insights into the origin of ligand bias in EGFR signaling. It has been argued that ligand bias in RTK signaling arises due to differential downregulation of the RTKs, or due to different abundances of cytoplasmic effectors[11,16]. Here we show that bias arises in the first step of signal transduction, along the length of the RTK.

We introduce the concept of mutation-induced bias coefficient, $\beta_{mut}$, which reports on the preferences of pathogenic RTK mutants to differentially phosphorylate tyrosines as compared to the wild-type RTKs. Mutation-induced bias is defined in analogy to ligand bias, where instead of comparing the effects of different ligands, we compare the wild-type and the mutant in the presence of the same ligand. By calculating both $\beta_{lig}$ and $\beta_{mut}$ from a comprehensive data set of dose-response curves, we uncouple and quantify biases introduced by ligand and by a pathogenic mutation. By simultaneously measuring the ligand binding and phosphorylation for a fluorescently labeled ligand, we quantify the characteristics of the transducer function, which ultimately allows the calculation of absolute bias coefficients for natural ligands.

RTK mutations have been mainly classified as either gain of function (activating) or loss of function (deactivating) mutations[40]. Here we show directly that the L834R EGFR mutation found in NSCLC induces bias in EGFR signal transduction across the plasma membrane. While EGFR signaling is biased toward Y1068 phosphorylation, the mutation switches the preference to Y1173 phosphorylation. It can be hypothesized that drug candidates that correct/unbias the first step in EGFR signal transduction can alter the signaling responses that are downstream from the mutant in a way that closely mimics WT EGFR signaling.

Our measurements set the stage for understanding how system bias modulates the effect of the L834R mutation on EGFR downstream signaling in physiological contexts. System bias acts in addition to ligand bias, and depends on the expression of downstream signaling molecules in the cells[6,15]. Measurements of EGFR ligand and mutation-induced bias in lung cancer cells will inform on the functional consequences of the differential signal transduction across the plasma membrane observed here. Studies can be expanded to investigate how the co-expression of WT EGFR and the L834R mutant affects signaling and cell physiology[41].

We hope that the demonstration of mutation-induced bias will create an impetus to quantify mutation-induced bias coefficients $\beta_{mut}$ for the many known RTK pathogenic mutations, and to reclassify the mutations based on the sign and magnitude of the bias coefficients. This will pave the way for the development of mutation-specific inhibitors which account for the discovered complexity in RTK signaling.

## Methods
### Plasmid constructs
The plasmid encoding for human EGFR, tagged with the fluorescent protein mTurquoise (mTurq) at the C-terminus via a flexible GGS linker,

is in the pSSX vector[42]. The L834R mutation was introduced in EGFR using the QuikChange II Site-Directed Mutagenesis Kit according to manufacturer's instructions (Agilent Technologies, #200523). Primers: 5′-cccagcagtttggcccgcccaaaatctgtga-3′ and 5′-tcacagattttgggcgggcca aactgctggg-3′. The plasmid used for the neural network training encoded for the extracellular and transmembrane domain (ECTM) of FGFR, a (GGS)5 linker, and mTurq in the pcDNA3.1(+) vector[43]. All plasmids were sequenced to confirm their identity (Genewiz).

## Cell culture and vesiculation
Chinese hamster ovary (CHO) cells were purchased from ATCC. CHO cells were used in these experiments as they do not exhibit endogenous EGFR expression[44]. CHO cells were cultured in Dulbecco's modified Eagle medium (Gibco, #31600034) supplemented with 10% fetal bovine serum (HyClone, #SH30070.03), 1 mM nonessential amino acids, 10 mM D-glucose, and 18 mM sodium bicarbonate at 37 °C in a 5% $CO_2$ environment. Cells were passed every other day using standard tissue culture techniques.

For vesiculation, the cells were seeded in a 6-well plate at a density of $2*10^4$ cells per well. 24 h later, the cells were transfected with 1 or 1.5 μg plasmid DNA using FuGene HD (Promega, #E2311) according to the manufacturer's protocol. 36 h after transfection, vesiculation was induced using osmotic stress as described in ref. 18. The osmotic pressure stresses the cells such that they release vesicles into solution without causing substantial cell detachment. The vesicles that were released in solution were collected by aspirating the supernatant with a cut 1000 μl micropipette tip.

## CHO vesicle characterization using dextran solutions
To characterize the permeability of CHO vesicles to macromolecules, FITC-labeled dextran was added to the vesicle solution and the ratio of FITC intensity inside and outside the vesicles was calculated for five different dextrans of sizes 20–2000 kDa. Dextrans were purchased from Sigma-Aldrich (#FD20S, #FD70S, #FD250S, #FD500S, and #FD2000S). After vesiculation, CHO vesicles with ECTM FGFR3 tagged with mTurq incorporated in the plasma membrane were transferred to an 8-well glass bottom chamber slide (ibidi, #80827). 100 nM of FITC-labeled dextran in osmotic chloride salt buffer was added to the vesicles. The chamber slide was transferred to a TCS SP8 confocal microscope (Leica Biosystems, Wetzlar, Germany) equipped with an automated stage and a HyD hybrid detector in photon counting mode. The vesicles were allowed to settle for 1 h. Image acquisition was automated by selecting pre-defined regions and focus points in the LAS X Navigator software (Leica Biosystems, Wetzlar, Germany). Two scans (256×256 pixels) per image were acquired, a 'mTurq'-scan (λ = 448 nm, emission window: 460–510 nm), where mTurq bound to FGFR3 is excited, and a 'FITC'-scan (λ = 488 nm, emission window: 500–540 nm), where FITC bound to dextran is excited. Images were acquired at 1% laser power with a 100 Hz scanning speed. To analyze the images, we developed a neural network approach as described in Supplementary Data.

## EGFR phosphorylation in CHO vesicles
To measure the phosphorylation of EGFR in the vesicle membranes, the vesicles were transferred to an 8-well glass bottom chamber slide (ibidi, #80827). The final concentrations of the ATP cocktail ingredients were 1 nM ATP, 10 mM $MgCl_2$, and 0.1 mM $Na_3VO_4$, a phosphatase inhibitor. The ligands used in this study were EGF (8916sf, Cell Signaling), TGFα (239A100, R&D Systems), Epiregulin (1195EP025 CF, R&D Systems), and EGF-tetramethylrhodamine (E3481, Thermofisher). For the transducer function measurements, we used commercially available EGF ligand from mouse that is labeled with rhodamine at its N-terminus (rho-mEGF, Thermofisher, E3481).

For detection of any phosphorylated Y residues, we used 67 nM of FITC-labeled anti-pY 4G10 antibody (05-321, Sigma Aldrich)[45]. For detection of Y1068 phosphorylation, 233 nM of AlexaF488-labeled anti-pY1068 EGFR antibody (IC3570G100, R&D Systems) was added[46]. Y1173 phosphorylation was detected using 50 nM AlexaF488-labeled anti-pY1173 EGFR antibody (NBP1-44893AF488, Novus Biologicals)[47].

Concentrations of the antibodies were chosen such that (i) the fluorescence intensities can be detected and measured on the plasma membrane (this depends on the labeling of the antibodies), (ii) the antibody amount exceeds the total amount of EGFR in the sample and (iii) the antibody concentration exceeds the anti-pY antibody dissociation constants (low nM). To determine the total EGFR concentration in a chamber slide well, 100 μl EGFR-mTurq vesicles were transferred to 96-well plates and full fluorescence emission spectra were collected with a H4 Synergy Hybrid Microplate Reader (BioTek Instruments, Winooski, VT). The samples were excited at 430 nm with a 9 nm bandwidth and the emitted fluorescence was collected from 450 to 620 nm with a 9 nm bandwidth with 5 nm steps. The emission spectra were corrected by subtracting the emission spectra of a vesicle sample derived from untransfected CHO cells. The maximum intensity of the corrected emission spectra at 475 nm was used to calculate the total EGFR concentration in a well upon calibration with purified solutions of mTurq of known concentration.

To monitor the reaction kinetics of EGFR phosphorylation, image acquisition was started right after the addition of the ligand/ATP cocktail to the vesicles. For dose response measurements, the phosphorylation reaction was allowed to reach equilibrium for 1 h prior to image acquisition. Image acquisition was automated by selecting predefined regions and focus points in the LAS X Navigator software (Leica Biosystems, Wetzlar, Germany). The reaction was monitored for up to 5 h. About 5000 images per experiment were acquired.

All images were acquired with a TCS SP8 confocal microscope (Leica Biosystems, Wetzlar, Germany) equipped with a motorized stage and a HyD hybrid detector in photon counting mode. Two scans per vesicle were taken, an 'mTurq'-scan (λ = 448 nm, emission window: 460–510 nm), where mTurq bound to EGFR is excited, and an 'AlexaF488'-scan (λ = 488 nm, emission window: 500–540 nm), where AlexaF488 bound to the anti phospho antibody is excited. The images (512×512 pixels) were acquired at 1% laser power with a 50 Hz scanning speed. Under these conditions, measured bleed-through coefficients were: mTurq in AlexaF488 channel <0.8%: AlexaF488 in mTurq channel <2.5%. These were considered negligible.

In experiments with unlabeled ligands, an mTurq/AlexaF488 FRET scan was performed: excitation: 448, emission: 500–540, to monitor if FRET occurs between mTurq and AlexaF488. Bleed through of mTurq into the FRET channel was 33%, and of AlexaF488 (due to direct excitation) was 7%. These values were used to determine the sensitized AlexaF488 fluorescence due to FRET between mTurq at the C-terminus of EGFR, and AlexaF488 on the antibody. FRET was negligible (Supplementary Fig. 2), and thus no correction for FRET was required.

In experiments with labeled ligand, the third scan was: excitation: 552, emission: 565–625, 3% laser power. The bleed-throughs of rhodamine in the mTurq and AlexaF488 channel were both <1.5% and were considered negligible.

## Ligand bias analysis
Dose response curves were fitted with the Hill equation with a slope of 1 (Supplementary Eq. 18), as prescribed for calculations of the bias coefficient $β_{lig}$[7,15,27,28]. The best fit $EC_{50}$ and $E_{top}$ were used to calculate $β_{lig}$ according to[7,12]:

$$\beta_{lig} = \log\left(\left(\frac{E_{top,A}EC_{50,B}}{EC_{50,A}E_{top,B}}\right)_{lig}\left(\frac{E_{top,B}EC_{50,A}}{EC_{50,B}E_{top,A}}\right)_{ref}\right) \tag{1}$$

where response A is Y1068 phosphorylation and response B is Y1173 phosphorylation.

The mutation-induced bias coefficient was calculated as:

$$\beta_{mut} = \log\left(\left(\frac{E_{top,A}EC_{50,B}}{EC_{50,A}E_{top,B}}\right)_{L834R}\left(\frac{E_{top,B}EC_{50,A}}{EC_{50,B}E_{top,A}}\right)_{WT}\right) \quad (2)$$

To test for ligand bias significance, a one-way ANOVA followed by Tukey's multiple comparisons test was performed using Graph-Pad Prism version 9.2.0 (GraphPad Software, San Diego, California USA).

The standard errors for bias coefficients and β′ values used in the statistical tests were derived from Monte-Carlo error estimations. For each parameter, $10^6$ normally distributed numbers were randomly generated using the mean and standard error of the parameter. The standard error of the distribution of the calculated bias coefficients was used for the statistical analysis.

### The transducer function

The transducer function relates a response to the stimulus that is causing it. Experimentally, signaling responses downstream of a receptor depend on the abundance of activated receptors through a hyperbolic dependence[6,48]. The hyperbolic dependence was derived from first principles by Black and Leff[30], and is the basis for their Operational Model, which is valid for different types of receptors including RTKs[12]. In this model, the ligand-bound receptors act as a stimulus that activates the response with an effective equilibrium dissociation constant denoted as $K_{resp}$′[49,50]:

$$\text{response} = \text{transducer function(stimulus)} = \frac{\text{stimulus}R_{max}}{\text{stimulus} + K_{resp}'} \quad (3)$$

In our experiments the "response" is the phosphorylation of a tyrosine in the intracellular domain of an RTK, $R_{phosho}$. We denote the maximum possible phosphorylation signal that can be achieved for this tyrosine in response to a full agonist as $R_{max}$[28,51].

The "stimulus" is the concentration of the ligand-bound receptors, [RL][30]. Therefore:

$$R_{phosho} = \frac{[RL]R_{max}}{[RL] + K_{resp}'} \quad (4)$$

If we divide both the numerator and denominator by the total receptor concentration, [Rt] and denote the fraction bound receptors, [RL]/[Rt], as $f_{bound}$, we obtain:

$$R_{phosho} = \frac{f_{bound}R_{max}}{f_{bound} + K_{resp}} \quad (5)$$

where

$$K_{resp} = \frac{K_{resp}'}{[Rt]} \quad (6)$$

and the "stimulus" is now redefined as the fraction of ligand-bound receptors, $f_{bound}$. $K_{resp}$ is the fraction of ligand-bound receptors that yields 50% of $R_{max}$. The value of $R_{max}$ depends on the fluorescent properties of antibodies used for the detection and thus the phosphorylation response is fully described by the ratio of $R_{phosho}/R_{max}$:

$$\frac{R_{phosho}}{R_{max}} = \frac{f_{bound}}{f_{bound} + K_{resp}} \quad (7)$$

The ligand-bound fraction $f_{bound}$ varies between 0 and 1. Setting $f_{bound} = 1$, we define:

$$\text{phosphorylation efficiency} = \frac{R_{phospho}(f_{bound}=1)}{R_{max}} = \frac{1}{1+K_{resp}} \quad (8)$$

This efficiency describes the maximum phosphorylation per receptor that can be achieved in response to a specific ligand, when all receptors are ligand-bound. It can be determined if the transducer function, given by Eq. (4), is measured experimentally and $R_{max}$ and $K_{resp}$ are determined from a two-parameter fit. The smaller the value of $K_{resp}$, the more efficient the phosphorylation. Phosphorylation efficiency of → 1 ($K_{resp}$ → 0) is indicative of a full agonist.

### The relation between $K_{resp}$ and bias, and the definition of absolute bias coefficients

Corrected dose-response curves are fitted using the Hill equation with $n = 1$ (Supplementary Eq. 19). The Black and Leff operational model is consistent with Supplementary Eq. 19, but also provides a physical-chemical description of the activation process[30]. According to the Black and Leff model, the concentration of the ligand-bound receptors [RL] in Eq. (4) depends on the concentrations of free receptor [R] and ligand [L], and on the effective ligand-receptor dissociation constant $K_L$ according to the equation:

$$[RL] = \frac{[R][L]}{K_L} \quad (9)$$

The total receptor concentration [Rt] is:

$$[Rt] = [R] + [RL] \quad (10)$$

Therefore:

$$[RL] = \frac{\frac{[Rt][L]}{K_L}}{1 + \frac{[L]}{K_L}} \quad (11)$$

Substitution of Eq. (11) into Eq. (4) yields:

$$\text{response} = \frac{[Rt][L]R_{max}}{[Rt][L] + K_{resp}'(K_L + [L])} = \frac{[Rt][L]R_{max}}{[L]([Rt] + K_{resp}') + (K_L K_{resp}')} \quad (12)$$

Dividing both the numerator and the denominator by $K_{resp}$′, we obtain:

$$\text{response} = \frac{\left(\frac{[Rt]}{K_{resp}'}\right)[L]R_{max}}{\left(\frac{[Rt]}{K_{resp}'} + 1\right)[L] + (K_L)} = \frac{\tau[L]R_{max}}{(\tau+1)[L] + (K_L)} \quad (13)$$

where $\tau$ is the "transducer coefficient" defined as:

$$\tau = \frac{[Rt]}{K_{resp}'} \quad (14)$$

Equation (13) can also be written as:

$$\text{response} = \frac{\tau/(\tau+1)[L]R_{max}}{[L] + (K_L)/(\tau+1)} \quad (15)$$

Now we see that Eq. (15) is the same as Supplementary Eq. 19, where

$$E_{top} = \frac{\tau R_{max}}{(\tau + 1)} \tag{16}$$

$$EC_{50} = \frac{K_L}{(\tau + 1)} \tag{17}$$

We use Eqs. (16) and (17) to arrive at an alternate expression of the bias coefficient

$$\beta_{lig} = \log\left(\left(\frac{E_{top,A}EC_{50,B}}{EC_{50,A}E_{top,B}}\right)_{lig}\left(\frac{E_{top,B}EC_{50,A}}{EC_{50,B}E_{top,A}}\right)_{ref}\right) \\ = \log\left(\left(\frac{\tau_A K_{L,B}}{K_{L,A}\tau_B}\right)_{lig}\left(\frac{\tau_B K_{L,A}}{K_{L,B}\tau_A}\right)_{ref}\right) \tag{18}$$

Assuming that the ligand binding coefficient $K_L$ does not depend on the binding of the anti-pY1068 and anti-pY1173 antibodies ($K_{L,A} = K_{L,B}$), we arrive at

$$\beta_{lig} = \log\left(\left(\frac{\tau_A}{\tau_B}\right)_{lig}\left(\frac{\tau_B}{\tau_A}\right)_{ref}\right) = \log\left(\left(\frac{\frac{[R_t]}{K_{resp',A}}}{\frac{[R_t]}{K_{resp',B}}}\right)_{lig}\left(\frac{\frac{[R_t]}{K_{resp',B}}}{\frac{[R_t]}{K_{resp',A}}}\right)_{ref}\right)$$
$$= \log\left(\left(\frac{K_{resp,B}}{K_{resp,A}}\right)_{lig}\left(\frac{K_{resp,A}}{K_{resp,B}}\right)_{ref}\right) \tag{19}$$
$$= \log\left(\frac{K_{resp,B}}{K_{resp,A}}\right)_{lig} - \log\left(\frac{K_{resp,A}}{K_{resp,B}}\right)_{ref} = \beta'^*_{lig} - \beta'^*_{ref}$$

Where we have used Eq. (6) and $\beta'^*_{lig}$ and $\beta'^*_{ref}$ are defined as:

$$\beta'^*_{lig} = \log\left(\frac{E_{top,A}EC_{50,B}}{EC_{50,A}E_{top,B}}\right)_{lig} = \log\left(\left(\frac{K_{resp,B}}{K_{resp,A}}\right)_{lig}\right)$$
$$= -\log\left(\left(\frac{K_{resp,A}}{K_{resp,B}}\right)_{lig}\right) \tag{20}$$

$$\beta'^*_{ref} = \log\left(\frac{E_{top,A}EC_{50,B}}{EC_{50,A}E_{top,B}}\right)_{ref} = \log\left(\left(\frac{K_{resp,B}}{K_{resp,A}}\right)_{ref}\right)$$
$$= -\log\left(\left(\frac{K_{resp,A}}{K_{resp,B}}\right)_{ref}\right) \tag{21}$$

This definition of $\beta'^*_{lig}$ and $\beta'^*_{ref}$ does not include measurement bias and these coefficients report on the preference for phosphorylation in absolute terms. If $K_{resp',B} > K_{resp',A}$, then the value of $\beta'^*$ is positive and response A is preferred. If $K_{resp',A} > K_{resp',B}$, then the value of $\beta'^*$ is negative and response B is preferred.

Once $\beta'^*_{ref}$ is known, we can calculate $\beta'^*_{lig}$ as:

$$\beta'^*_{lig} = \beta_{lig} - \beta'^*_{ref} \tag{22}$$

By analogy, we can calculate the absolute bias coefficient $\beta'^*_{mut}$ as:

$$\beta'^*_{mut} = \beta_{mut} - \beta'^*_{ref} \tag{23}$$

## Reporting summary
Further information on research design is available in the Nature Portfolio Reporting Summary linked to this article.

## Data availability
All single vesicle data generated in this study have been deposited in the figshare database under accession code: https://figshare.com/articles/dataset/Quantification_of_ligand_and_mutation-induced_bias_in_EGFR_phosphorylation_in_direct_response_to_ligand_binding/24162846. Source data are provided with this paper.

## Code availability
The vesicle analysis code has been deposited under accession code: https://gitlab.com/hristovagroup/vesicle-analysis. The code to correct for unliganded EGFR dimers has been deposited under accession code: https://gitlab.com/hristovagroup/unliganded-dimer-correction/-/tree/main.

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

## Acknowledgements

Supported by NIH GM068619 (K.H.) and NSF MCB 2106031 (K.H.). We thank Dr. M.D. Paul for helpful discussions and thermodynamic model development.

## Author contributions

D.W. designed and performed the experiments. D.W and E.O analyzed the data. D.W. wrote the first draft of the paper. K.H. designed the experiments and secured funding. All authors edited the paper.

## Competing interests

The authors declare no competing interests.
