## [Peer Review File · Nature Communications]

Quantification of ligand and mutation-induced bias in EGFR phosphorylation in direct response to ligand bindingReviewers' comments:

Reviewer #1 (Remarks to the Author):

The manuscript titled “Quantification of ligand and mutation-induced bias in EGFR phosphorylation in direct response to ligand binding” uses isolated plasma membrane vesicles, automatic image processing, and neural network learning to quantitate the binding of several ligands and phosphorylation states of EGFR. The authors state that the plasma membrane vesicles represent the optimal state in which to measure ligand bias due to their inherent permeability, which allows large molecules such as antibodies to pass through and cytoplasmic contents to dilute out. While this idea is creative and seeks to answer a specific problem related to ligand bias of various receptor tyrosine kinases, I find that the current manifestation is lacking.

The plasma membrane vesicles that are used in this study are not representative of the intact plasma membrane of cells. The lipid compositions that are referenced in ref 19 are descriptive at best and do not actually show that osmotically derived vesicles from CHO cells are similar to the other preparation method (utilizing formaldehyde and DTT). The authors state that the cells from which the vesicles are derived from undergo sustained stress for >12hr. What this stress does to lipid and protein expression is not shown here and needs to be shown for the massive assumptions to be valid.

Furthermore, as stated in the referenced works, the vesicles have lost the asymmetry of lipids that defines intact plasma membranes. This scrambling of lipids results in dramatically different biophysical and compositional environments in which the receptors are solvated. If the receptors have any electrostatic or specific interactions with cytoplasmic-facing lipids such as phosphatidylserine or phosphorylated phosphatidylinositols, these will be lost or diminished in these vesicles, either by scrambling or by phosphatases or lipases which are activated in the vesiculation process. Additionally, the biophysical environment (e.g. diffusion, lipid packing, etc) is altered in scrambled membranes, and the biophysical environment has been shown to have an effect on transmembrane receptor structure and function. In saying this, I applaud the authors' creative approach to minimize the additional contaminating effects of feedback response in intact cells, but I do not feel that the quantitative kinetic results given here will necessarily represent actual values found in cells.

Additionally, I understand that the authors chose CHO cells so that they could control the EGFR expression in their vesicles; however, the lipid compositions of various cells are quite different. Thus, it's unclear whether the artificial expression in these cells is representative of actual EGFR localization, signaling, and behavior that might be seen in cells endogenously expressing EGFR.

Additional comments

1. The abstract doesn't mention the results of the findings nor the potentially artifactual technique used exclusively in this work. Instead, it presents bold ideas that aren't actually shown here.
2. The methods section is lacking in detail. Much of what is stated in the supplement should be stated instead in the methods section.

3. It would be informative to do the same experiment in intact cells and see the potential feedback that alters the kinetics measured in vesicles and show here as a comparison. An additional experimental setup, a different RTK, and/or different cells should be used to validate the results and experiments done here.
4. The curves in S7 or S8 should be shown in the manuscript, not supplement.
5. The figures are difficult to read. The graphs axes fonts are tiny; the data points are tiny; yet the phospho-Tyrosine or mutant font is huge.
6. I'm not sure why the authors needed to measure >1000 vesicles per condition nor why neural network learning was necessary. This was never made clear in the supplement nor discussion. If this is important to validate the results, then a figure should be dedicated to the calibration and justification of this technique.

Reviewer #2 (Remarks to the Author):

In this article, the authors introduce a new methodology to tease apart signalling effects arising from ligand vs. system bias in response to different EGFR natural agonists. They later apply their approach to characterise signalling bias arising from a common mutation of the receptor found in non-small-cell lung cancer (NSCLC) patients. Regarding their findings on the wild type receptor, the fact they observe bias between agonists in the absence of other potential confounding factors speaks of the possibility to design or screen for new EGFR ligands that shift EGFR tyrosine phosphorylation patterns and thus signalling responses. In the case of the mutated receptor, however, their experimental system may not actually reflect the signalling response to these ligands found in most cases of NSCLC, where we find heterozygous EGFR genes (i.e., cancer cells express a mutated and a wild type receptor version). In more general terms, the experimental approach used in this work makes it difficult to assess if the differential tyrosine phosphorylation observed in response to different endogenous ligands would finally translate into significant variation in signalling effects, a limitation that should at least be stated in the discussion part of the manuscript.

Major comments:

1. Although determining how cancer-associated mutations like EGFR L834R can bias responses to endogenous ligands can be of interest and suggest new strategies to find new molecules counteracting pathogenic bias, the way this has been measured in the manuscript may not reflect these responses closely. If we consider existing evidence on the L834R mutation in NSCLC, most reports have detected it to be a heterozygous somatic mutation, with one mutated and one wild type copy of the EGFR gene in found lung cancer cells. Therefore, as RTKs function as obligate dimers and considering heterodimers between both receptor versions are known to form and may have different activation properties than homomers of wild type or mutated receptors (as summarised in PMID: 24023066), the authors should have measured tyrosine phosphorylation in response to the three endogenous ligands in the presence of both wild type and mutated receptors and not only for mutated receptors in isolation.

2. In terms of nomenclature, I am not sure that the term intrinsic in intrinsic ligand bias is required, as the ligand bias the authors are measuring already excludes system bias (which, based on the work on signalling bias in the GPCR field, would also include the effects of intracellular partners participating in feedback loops). The same happens with the intrinsic signalling bias induced by the cancer mutation, where altered signalling arising from a pathogenic mutation is difficult to justify as an intrinsic receptor factor. Usually bias at the receptor level (e.g., the one found in 7 transmembrane receptors that only naturally couple to arrestins or that arising from mutations / polymorphisms) is classed as receptor bias. Alternatively, the authors should consider consistently using mutation-induced bias, a term they mention later in the manuscript, as this seems very pertinent and explanatory for the kind of bias they measure.

Additional comments:

a) I agree that the experiments presented in this manuscript provide evidence that bias among endogenous EGFR agonists is, at least partly, driven by the stabilization of different receptor conformational states by ligands, a phenomenon that would parallel observations in other receptors like GPCRs. This, in turn, opens new opportunities for therapeutic intervention through the design of or search for biased receptor modulators as the authors rightly claim. However, in terms of translational research and the impact of their findings, the fact that RTK activation will depend on the differential expression of downstream effectors and modulators cannot be overlooked. This is especially true in the context of cancer cells, where we can expect dramatic shifts in protein abundance and where system bias may be more pronounced than when we compare signalling outcomes between healthy cell types or tissues for other receptors like GPCRs. Therefore, the authors should at least discuss in the last section how their findings could be further explored in more complex experimental settings that better reproduce the whole EGFR signalling process in physiological and pathological conditions.

b) The authors need to clarify some comments on how their results match previous observations from the literature. In page 7, they say: EGF and epiregulin are biased ligands, consistent with reports of ligand functional selectivity in cells. Considering this manuscript claims epiregulin is biased when compared to EGF and TGF α , but the latter ligands are not biased with respect to one another, how did previous work reach that conclusion? Did they use an alternative endogenous ligand that could have been also included in this analysis as a reference? Or do the authors mean that EGF and epiregulin are biased with respect to each other? In any case, the current wording does not clarify the statement that: The data acquired with the new method can be compared to published data.

Reviewer #3 (Remarks to the Author):

The authors analyze EGFR activation in osmotically-generated, cell derived vesicles and compare the potential of different ligands to induce Y1068 and Y1173 phosphorylation in the wild-type and the L834R mutant. While the bias analysis and determination of the transducer function allow some more direct, visual insight into the workings of the receptor, it does not significantly go beyond what is possible by

comparing phosphorylation levels and EC50 values for the phosphorylation of different tyrosines. Furthermore, there are several issues with the approach of the quantitative analysis.

1. The authors use a neural network-based object detection to identify “good vesicles”. The approach seems to work properly according to Fig. S1a,b, but presentation of a confusion matrix is a more widely accepted tool to display the performance of the object detection. Furthermore, the authors seem to analyze all pixels within the bounding box of the object, which includes an overwhelming majority of background pixels, and it may also include the edge of other vesicles, as exemplified in the upper right corner of Fig. S1a. Pixel classification (semantic segmentation) should have been performed with the identified “good vesicles” to analyze only the pixels of interest, i.e., the membrane pixels.
2. The authors fit the responses with a Hill equation with the cooperativity coefficient equal to 1 (equation S18). The EGFR system is known to exhibit cooperativity, either positive or negative depending on the experimental conditions. The consequence of this simplification has not been taken into consideration. This issue is especially relevant since the dose-response curves (Fig 2A, 3A, S6) seem to display markedly different apparent cooperativities judged by the concentration range between the 10% response and 90% response.
3. When imaging the vesicles labeled with three fluorescent dyes (bottom of p5), no correction is performed for spectral spillover and the potential FRET between mTurquoise and Alexa488 is also neglected.
4. Correction for the contribution of unliganded dimers according to the scheme presented in Fig. S4 is rather simple considering structural and modeling studies showing that a much higher number of species of EGFR are present. Furthermore, the authors used model parameters from previous publications determined for cells. It is quite likely that these parameters are influenced by the experimental conditions, i.e., the fact that vesicles are used in the current study. In light of these shortcomings, the correction is deemed rather arbitrary. Since the corrected dose-response curves are provided by this model, and the EC50 and Etop values in the tables (e.g., in Fig. 2) are calculated from the corrected dose-response curves, the potential effect of this simplification may be significant.
5. The fraction of phosphorylated tyrosines reported in Fig. 2A and 3A was determined by dividing the Alexa488 intensity of the anti-pY antibody with the intensity of mTurquoise of the receptor. Fluorescence intensity depends on the excitation laser intensities, the molar absorption coefficients, the quantum yields of the fluorophores, the number of fluorophores / antibody (degree of labeling) and the detection efficiencies of the detector system. Therefore, simply dividing the two intensities by each other will not correspond to the fraction of receptors in which a certain tyrosine is phosphorylated, but will be a relative measure of the phosphorylation level. For this reason, the fact whether a ligand is a full or partial agonist cannot be concluded from this data.
6. f_{bound} is significantly larger than 1 for many vesicles in Fig. 5C although its theoretical maximum is 1. The SD of the phosphorylations shown in Figs. 2A and 3A is extremely large. Such problems may arise from the methodological shortcomings presented above.

Reviewer #1

The plasma membrane vesicles that are used in this study are not representative of the intact plasma membrane of cells.

OUR RESPONSE. We completely agree with all the differences that were pointed out by the reviewer. We developed the model system of plasma membrane derived vesicles as a better alternative of the nanodisks used to study membrane proteins, and now we leverage it to study RTK phosphorylation. The goal here was to gain biophysical knowledge that cannot be gained through cellular studies. We were inspired by the remarkable successes in understanding mechanisms of GPCR signaling through the use of nanodisks. In the nanodisks, GPCRs can bind to and activate G proteins or arrestins on the cytoplasmic side, in response to ligand-binding on the extracellular side (i.e. both the extracellular and intracellular side of the receptor are accessible (4, 5)). We wanted to develop a model system that can be used for RTKs, and is more biologically relevant than the nanodisks, which just incorporate a few lipid types.

Nanodisks are not appropriate for studies of RTKs, in two important ways. First, the nanodisk puts artificial constraints on the oligomer size of the RTK assemblies, which can be dimers or higher-order oligomers. Second, while it is trivial to purify a GPCR and then reconstitute in lipid bilayers it without losing activity, this is a formidable challenge for RTKs and other single pass receptors. We wanted to establish a model system that does not require RTK extraction out of the native plasma membrane and provides a contiguous membrane to ensure that the RTKs can associate with each other as they do in cells.

Notably, plasma membrane derived vesicles can also be used to study GPCR signaling. We believe that this model system has advantages over nanodisks as it is much closer to the native membrane environment in which the GPCRs reside.

Finally, it is worth noting that dissociation constants describing EGFR homodimerization have been measured by us in vesicles and by Linda Pike in cells. Both measurements yield the same results, supporting the relevance of the model system for EGFR studies.

1. The abstract doesn't mention the results of the findings nor the potentially artifactual technique used exclusively in this work. Instead, it presents bold ideas that aren't actually shown here.

OUR RESPONSE. We have re-written the abstract.

2. The methods section is lacking in detail. Much of what is stated in the supplement should be stated instead in the methods section.

OUR RESPONSE. We have expanded the methods section. Based on this comment, and comments from reviewer 3, we have moved all equations pertaining to the transducer function, the phosphorylation efficiency, and the absolute bias coefficients to the main text. These are all novel concepts introduced here and measured for RTKs for the first time in this work.

3. It would be informative to do the same experiment in intact cells and see the potential feedback that alters the kinetics measured in vesicles and show here as a comparison. An additional experimental setup, a different RTK, and/or different cells should be used to validate the results and experiments done here.

OUR RESPONSE. The type of kinetics experiments mentioned by the reviewer have been performed many times in the literature, and the relevant citations are included in the revised manuscript. The kinetics traces invariably show an increase in phosphorylation over time, followed by a decrease due to signal attenuation in cells (due to feed-back loops). We do not believe that such kinetics traces can verify the conclusions of this study.

To address the reviewer's concerns, we have elaborated on results in the literature that support our finding that EGF and epiregulin are biased ligands. Ligand bias leads to fundamentally different biological outcomes. It has been shown in the literature that epiregulin induces differentiation under the same conditions where EGF induces proliferation, suggesting that the two are biased ligands in support of our conclusions (10).

4. The curves in S7 or S8 should be shown in the manuscript, not supplement.

OUR RESPONSE. We believe that these figures are not critical to the conclusion of the paper, and we have not moved them in this revision. To arrive at this decision, we took into account comments by reviewer 3.

5. The figures are difficult to read. The graphs axes fonts are tiny; the data points are tiny; yet the phospho-Tyrosine or mutant font is huge.

OUR RESPONSE. We have edited the figures

6. I'm not sure why the authors needed to measure >1000 vesicles per condition nor why neural network learning was necessary. This was never made clear in the supplement nor discussion. If this is important to validate the results, then a figure should be dedicated to the calibration and justification of this technique.

OUR RESPONSE. We have clarified this issue in the revised manuscript, and we have included the relevant citation. We have shown previously that the “scatter” in single vesicle fluorescence data is due to white noise in the measurements (11). In the case of white noise, errors are minimized by performing many measurements. For this work, we need to know the averages very well (see last comment of reviewer 3).

Reviewer #2 (Remarks to the Author):

1. Although determining how cancer-associated mutations like EGFR L834R can bias responses to endogenous ligands can be of interest and suggest new strategies to find new molecules counteracting pathogenic bias, the way this has been measured in the manuscript may not reflect these responses closely. If we consider existing evidence on the L834R mutation in NSCLC, most reports have detected it to be a heterozygous somatic mutation, with one mutated and one wild type copy of the EGFR gene in found lung cancer cells. Therefore, as RTKs function as obligate dimers and considering heterodimers between both receptor versions are known to form and may have different activation properties than homomers of wild type or mutated receptors (as summarised in PMID: 24023066), the authors should have measured tyrosine phosphorylation in response to the three endogenous ligands in the presence of both wild type and mutated receptors and not only for mutated receptors in isolation.

OUR RESPONSE. We have cited the paper the reviewer is referring to, and we have stated the importance of heterodimerization in the Discussion of the revised paper. Note, however, that WT/MUT heterodimers will not exist in isolation. They will form alongside WT/WT and MUT/MUT homodimers, since the homo- and heterodimerization equilibria are coupled for the RTKs. The relative abundancies will depend on the expression levels and on the three relevant dissociation constants.

We have been studying heterointeractions for many years (12-14), and we have developed methods to measure both homodimerization and heterodimerization dissociation constants. To perform the experiment that the reviewer is suggesting, first we have to measure the WT_EGFR/L834R_EGFR heterodimerization constant, so we know how many heterodimers we have when we measure phosphorylation. Thus far, we have only measured the two homodimerization constants for WT EGFR and L834R_EGFR (6). We are looking forward to making these measurements, and to developing new methodologies to understand the signaling of the heterodimers. The latter will be only possible once the WT and MUT receptors are fully characterized in terms of their efficacy, potency, and bias (as in the current work). The difficult

heterodimer project is feasible as we will build upon the many quantitative fluorescence methods that we have established.

2. In terms of nomenclature, I am not sure that the term intrinsic in intrinsic ligand bias is required, as the ligand bias the authors are measuring already excludes system bias (which, based on the work on signalling bias in the GPCR field, would also include the effects of intracellular partners participating in feedback loops). The same happens with the intrinsic signalling bias induced by the cancer mutation, where altered signalling arising from a pathogenic mutation is difficult to justify as an intrinsic receptor factor. Usually bias at the receptor level (e.g., the one found in 7 transmembrane receptors that only naturally couple to arrestins or that arising from mutations / polymorphisms) is classed as receptor bias. Alternatively, the authors should consider consistently using mutation-induced bias, a term they mention later in the manuscript, as this seems very pertinent and explanatory for the kind of bias they measure.

OUR RESPONSE. We understand the arguments of the reviewer. The methods that we use, i.e. the construction of the bias plots and the calculations of bias coefficients are specifically developed in order to filter out the contributions of system bias and measurement bias (15-17), such that the ligand bias can be actually quantified. However, a recent review on best practices of bias identification highlights many outstanding challenges in the field and contradicting results in the literature (15). As such we decided to use “intrinsic ligand bias” to indicate that there is no system bias contribution in our measurements. We think that this is well justified. However, we have removed the word “intrinsic” when we refer to mutation-induced bias, to address the concern of the reviewer.

Additional comments:

a) I agree that the experiments presented in this manuscript provide evidence that bias among endogenous EGFR agonists is, at least partly, driven by the stabilization of different receptor conformational states by ligands, a phenomenon that would parallel observations in other receptors like GPCRs. This, in turn, opens new opportunities for therapeutic intervention through the design of or search for biased receptor modulators as the authors rightly claim. However, in terms of translational research and the impact of their findings, the fact that RTK activation will depend on the differential expression of downstream effectors and modulators cannot be overlooked. This is especially true in the context of cancer cells, where we can expect dramatic shifts in protein abundance and where system bias may be more pronounced than when we compare signalling outcomes between healthy cell types or tissues for other receptors like GPCRs. Therefore, the authors should at least discuss in the last section how their findings could be further explored in more complex experimental settings that better reproduce the whole EGFR signalling process in physiological and pathological conditions.

OUR RESPONSE. We have expanded the text, as suggested, to discuss system bias. Now that we have these measurements, the stage is set for understanding the contributions of system bias

to the effect of the L834R mutation on cell physiology. System bias acts in addition to ligand bias, and depends on the expression of downstream signaling molecules in the cells.

b) The authors need to clarify some comments on how their results match previous observations from the literature. In page 7, they say: EGF and epiregulin are biased ligands, consistent with reports of ligand functional selectivity in cells. In any case, the current wording does not clarify the statement that: The data acquired with the new method can be compared to published data.

OUR RESPONSE. We have elaborated on results in the literature that support our finding that EGF and epiregulin are biased ligands. Ligand bias leads to fundamentally different biological outcomes. It has been shown in the literature that epiregulin induces differentiation under the same conditions where EGF induces proliferation, suggesting that the two are biased ligands in support of our conclusions.

Considering this manuscript claims epiregulin is biased when compared to EGF and TGF α , but the latter ligands are not biased with respect to one another, how did previous work reach that conclusion? Did they use an alternative endogenous ligand that could have been also included in this analysis as a reference? Or do the authors mean that EGF and epiregulin are biased with respect to each other?

OUR RESPONSE. For the three ligands, we perform three pairwise comparisons: TGF α versus EGF, epiregulin vs EGF, and epiregulin vs TGF α . For each comparison, one of the ligands is chosen as a reference ligand. We have emphasized this in the revised manuscript.

Reviewer #3 (Remarks to the Author):

1. The authors use a neural network-based object detection to identify “good vesicles”. The approach seems to work properly according to Fig. S1a,b, but presentation of a confusion matrix is a more widely accepted tool to display the performance of the object detection. Furthermore, the authors seem to analyze all pixels within the bounding box of the object, which includes an overwhelming majority of background pixels, and it may also include the edge of other vesicles, as exemplified in the upper right corner of Fig. S1a. Pixel classification (semantic segmentation) should have been performed with the identified “good vesicles” to analyze only the pixels of interest, i.e., the membrane pixels.

OUR RESPONSE. We believe that the reviewer was left under the impression that the neural network program performs the quantification of vesicle membrane intensities. This is not so, and it is our fault that not all steps were described appropriately. We have re-written the text to clarify this, and we have added a new figure. The neural network is only the first step in the quantification of vesicle intensities. Its role is to pre-screen the microscope images and to crop out sub-images containing “good vesicles” for further analysis, very much like a researcher who is manually focusing on a vesicle in the microscope to take the vesicle image, and then cropping

the vesicle image to feed it into the vesicle analysis program. These sub-images are then imported into a Matlab code, which has been used extensively for FRET vesicle data analysis (1-3, 6, 9, 11, 18). This code identifies the center of the vesicles and the pixels belonging to the membrane, and calculates intensities in the plasma membrane (see Figure S1B). The code rejects vesicles which are not perfectly spherical or have other defects.

Please note, we have deposited the relevant codes so the reviewer can inspect them. They can be found at <https://gitlab.com/hristovagroup/vesicle-analysis>

2. The authors fit the responses with a Hill equation with the cooperativity coefficient equal to 1 (equation S18). The EGFR system is known to exhibit cooperativity, either positive or negative depending on the experimental conditions. The consequence of this simplification has not been taken into consideration. This issue is especially relevant since the dose-response curves (Fig 2A, 3A, S6) seem to display markedly different apparent cooperativities judged by the concentration range between the 10% response and 90% response.

OUR RESPONSE. We agree with the reviewer, and we are careful to not over-interpret the bias coefficient calculations. Along with bias coefficients, we show bias plots which are constructed using uncorrected experimental data, and we draw conclusions based on these bias plots. In the literature, the bias plots are considered the most reliable proof of bias (16, 17), while bias coefficients are calculated to support the bias plots as they allow statistical analysis (p value calculations). The bias plot provide a visual demonstration of bias. In our case, the visual comparison is informative because the standard errors are very small (as the white noise in image collection is minimized with our imaging protocols).

Further, we want to clarify that:

1. In this first study of ligand bias for an RTK, we have closely followed protocols that pharmacologists use and recommend to assess ligand bias (15-17). The calculation of the bias coefficient β_{lig} **requires** that the Hill equation is used with n fixed at 1. If the fit is performed with variable n, the equation for β_{lig} cannot be used because it is derived for n=1 only. We are currently working on a method to calculate a novel type of bias coefficient (we call it “kappa”) which does not require that n=1. This paper is under review. However, we think that it is prudent in the current work to use methods that have been used for decades and are recommended in the pharmacology literature. As such, we present bias plots, and we calculate β_{lig} to support results from the bias plot. The bias plots and the calculated bias coefficients are in agreement.
2. We agree that the EGFR system exhibits cooperativity in binding. However, here we measure the phosphorylation response, not the binding. The phosphorylation depends on the value of K_{resp} (see equation 5), and cannot be predicted based on binding only.

3. When imaging the vesicles labeled with three fluorescent dyes (bottom of p5), no correction is

performed for spectral spillover and the potential FRET between mTurquoise and Alexa488 is also neglected.

OUR RESPONSE. We have expanded the method section with experimental details that address the reviewer's concerns. Spectral spill-over (or bleed-through) is always measured, and so is the FRET. This information is now provided in the Supplement (see Fig S2). In this study, the bleed-throughs and the FRET turned out to be negligible, so we did not have to correct for them.

To best demonstrate zero FRET, we have included kinetics traces in three channels in Figure S2. The sensitized acceptor emission is zero. The donor fluorescence in the presence and in the absence of the acceptor is the same, indicative of zero FRET.

4. Correction for the contribution of unliganded dimers according to the scheme presented in Fig. S4 is rather simple considering structural and modeling studies showing that a much higher number of species of EGFR are present. Furthermore, the authors used model parameters from previous publications determined for cells. It is quite likely that these parameters are influenced by the experimental conditions, i.e., the fact that vesicles are used in the current study. In light of these shortcomings, the correction is deemed rather arbitrary. Since the corrected dose-response curves are provided by this model, and the EC50 and Etop values in the tables (e.g., in Fig. 2) are calculated from the corrected dose-response curves, the potential effect of this simplification may be significant.

OUR RESPONSE. First, we have expanded the text to clarify that the measurements of association constant for EGFR are the same in cells and in vesicles. Second, we have expanded the discussion to emphasize that the bias plots, which provide a visual demonstration of bias, are constructed from the uncorrected data. Third, we have emphasized that the bias plots are considered the most reliable proof of bias in the literature. Fourth, we have discussed the assumptions for the bias coefficient calculations, and we have discussed that the calculated bias coefficients support the conclusions from the bias plots.

5. The fraction of phosphorylated tyrosines reported in Fig. 2A and 3A was determined by dividing the Alexa488 intensity of the anti-pY antibody with the intensity of mTurquoise of the receptor. Fluorescence intensity depends on the excitation laser intensities, the molar absorption coefficients, the quantum yields of the fluorophores, the number of fluorophores / antibody (degree of labeling) and the detection efficiencies of the detector system. Therefore, simply dividing the two intensities by each other will not correspond to the fraction of receptors in which a certain tyrosine is phosphorylated, but will be a relative measure of the phosphorylation level. For this reason, the fact whether a ligand is a full or partial agonist cannot be concluded from this data.

OUR RESPONSE. What the reviewer is saying is exactly correct. This is why typically no one knows whether a so called "full agonist" is really a full agonist. As our colleagues working with GPCRs say: "you have a full agonist until somebody else finds a better agonist". But we have overcome this limitation, for the first time. We have measured the transducer function of RTK phosphorylation for the first time, which allowed us to calculate the phosphorylation efficiency. This novel development is based on a theoretical framework based on the operational model of

Black and Leff (19), which we have implemented in this study. We have moved the relevant equations out of the supplement to the main text, and we have expanded the Discussion to better emphasize the novelty of the work.

6. f_{bound} is significantly larger than 1 for many vesicles in Fig. 5C although its theoretical maximum is 1. The SD of the phosphorylations shown in Figs. 2A and 3A is extremely large. Such problems may arise from the methodological shortcomings presented above.

OUR RESPONSE. No, we set the **average** phosphorylation at high ligand concentration (the plateau) to 1.

The errors associated with imaging of single vesicles are due to white noise (11). Thus, it is incorrect to set the measurement for a single vesicle to 1. If the errors are due to white noise, they are minimized by collecting a large number of data points (which we do), so it is the **average** which is meaningful. The distribution of data is normal, the width is due to white noise, the errors of the means are very small, and the measurements are of high precision.

1. Chen, L., J. Placone, L. Novicky, and K. Hristova. 2010. The extracellular domain of fibroblast growth factor receptor 3 inhibits ligand-independent dimerization. *Science Signaling* 3:ra86.
2. Sarabipour, S., and K. Hristova. 2016. Mechanism of FGF receptor dimerization and activation. *Nat. Commun.* 7:10262.
3. Sarabipour, S., K. Ballmer-Hofer, and K. Hristova. 2016. VEGFR-2 conformational switch in response to ligand binding. *Elife* 5.
4. Choi, M., D. P. Staus, L. M. Wingler, S. Ahn, B. Pani, W. D. Capel, and R. J. Lefkowitz. 2018. G protein-coupled receptor kinases (GRKs) orchestrate biased agonism at the beta2-adrenergic receptor. *Sci Signal* 11.
5. Zhou, X. E., K. Melcher, and H. E. Xu. 2017. Understanding the GPCR biased signaling through G protein and arrestin complex structures. *Curr Opin Struct Biol* 45:150-159.
6. Byrne, P. O., K. Hristova, and D. J. Leahy. 2020. EGFR forms ligand-independent oligomers that are distinct from the active state. *The Journal of biological chemistry* 295:13353-13362.
7. Macdonald, J. L., and L. J. Pike. 2008. Heterogeneity in EGF-binding affinities arises from negative cooperativity in an aggregating system. *Proceedings of the National Academy of Sciences of the United States of America* 105:112-117.
8. Tao, X., C. Zhao, and R. MacKinnon. 2023. Membrane protein isolation and structure determination in cell-derived membrane vesicles. *Proc Natl Acad Sci U S A* 120:e2302325120.
9. Kavran, J. M., J. M. McCabe, P. O. Byrne, M. K. Connacher, Z. H. Wang, A. Ramek, S. Sarabipour, Y. B. Shan, D. E. Shaw, K. Hristova, P. A. Cole, and D. J. Leahy. 2014. How IGF-1 Activates its Receptor. *Elife* 3.
10. Freed, D. M., N. J. Bessman, A. Kiyatkin, E. Salazar-Cavazos, P. O. Byrne, J. O. Moore, C. C. Valley, K. M. Ferguson, D. J. Leahy, D. S. Lidke, and M. A. Lemmon. 2017.

- EGFR Ligands Differentially Stabilize Receptor Dimers to Specify Signaling Kinetics. *Cell* 171:683-695 e618.
11. Chen, L. R., L. Novicky, M. Merzlyakov, T. Hristov, and K. Hristova. 2010. Measuring the Energetics of Membrane Protein Dimerization in Mammalian Membranes. *Journal of the American Chemical Society* 132:3628-3635.
 12. Del Piccolo, N., S. Sarabipour, and K. Hristova. 2017. A New Method to Study Heterodimerization of Membrane Proteins and Its Application to Fibroblast Growth Factor Receptors. *The Journal of biological chemistry* 292:1288-1301.
 13. Paul, M. D., H. N. Grubb, and K. Hristova. 2020. Quantifying the strength of heterointeractions among receptor tyrosine kinases from different subfamilies: Implications for cell signaling. *The Journal of biological chemistry* 295:9917-9933.
 14. Paul, M. D., and K. Hristova. 2021. Interactions between Ligand-Bound EGFR and VEGFR2. *J Mol Biol* 433:167006.
 15. Kolb, P., T. Kenakin, S. P. H. Alexander, M. Bermudez, L. M. Bohn, C. S. Breinholt, M. Bouvier, S. J. Hill, E. Kostenis, K. A. Martemyanov, R. R. Neubig, H. O. Onaran, S. Rajagopal, B. L. Roth, J. Selent, A. K. Shukla, M. E. Sommer, and D. E. Gloriam. 2022. Community guidelines for GPCR ligand bias: IUPHAR review 32. *Br J Pharmacol* 179:3651-3674.
 16. Kenakin, T. 2019. Biased Receptor Signaling in Drug Discovery. *Pharmacol Rev* 71:267-315.
 17. Kenakin, T. 2017. Signaling bias in drug discovery. *Expert Opin Drug Discov* 12:321-333.
 18. Del Piccolo, N., J. Placone, and K. Hristova. 2015. Effect of Thanatophoric Dysplasia Type I Mutations on FGFR3 Dimerization. *Biophysical Journal* 108:272-278.
 19. Black, J. W., and P. Leff. 1983. Operational models of pharmacological agonism. *Proc R Soc Lond B Biol Sci* 220:141-162.

REVIEWER COMMENTS

Reviewer #1 (Remarks to the Author):

The manuscript titled “Quantification of ligand and mutation-induced bias in EGFR phosphorylation in direct response to ligand binding” uses isolated plasma membrane vesicles, automatic image processing, and neural network learning to quantitate the ligand bias towards different phosphorylation states of EGFR. The authors use plasma membrane vesicles which are a more representative form of a native plasma membranes than reconstitution studies or nanodisks. The authors have made a decent number of changes suggested by reviewer comments to make the manuscript better. It is now cleaner, more readable, and easier to follow with more experimental details. As this is in part a methodological advancement, I do think that more information regarding the use of the isolated osmotically produced vesicles needs to be added for clarity and accuracy of the methodology.

As I mentioned previously, and this was not addressed in this resubmission, the plasma membrane vesicles that are used in this study are not representative of the intact plasma membrane of cells, nor is there any data shown in this manuscript suggesting that they would be. The authors state: “Previous work has shown that the lipid composition is very similar to the lipid composition of the plasma membrane (19).” In the referenced work, the authors do not actually compare the lipid composition of these osmotically produced vesicles to the plasma membrane of cells, but rather to chemically produced plasma membrane vesicles. And even then, the referenced work highlights statistically significant changes between those two preparations. Specifically, the authors need to expand on their statement in the discussion: “These vesicles preserve many, although not all, characteristics of the plasma membrane.” The authors did add a statement discussing the loss of the cytoskeletal attachment and asymmetry of lipids, which is an improvement, but an elaboration of what is meant by which characteristics of these vesicles are similar to cellular plasma membranes should be also given.

I appreciate the authors’ comparison of these vesicles to nanodisks and suggest that what was written in the reviewer response be added to the discussion in the manuscript. I think the valid comparison here for this work is vesicles to nanodisks, not vesicles to intact, native plasma membranes of cells, and more emphasis throughout the work should be on this.

Reviewer #3 (Remarks to the Author):

A. When reading the first version of the manuscript, I raised issues about image segmentation (point 1) and the lack of correction for overspill and FRET (point 3). The authors convincingly rebutted this criticism in their response, and modified the manuscript accordingly. I also accept that bias itself can be seen from the bias plots that are generated from raw data.

B. My other major concerns revolved around how realistic the calculations are given that they admittedly

ignore significant aspects of EGFR dimerization. The fact that the calculations allowed the authors to reach the same conclusions as those based on bias plots does not verify the validity of the calculations themselves. While the authors attempt not to overinterpret them, still a significant fraction of the manuscript is devoted to these calculations. Specifically, limitation of the model used for correcting for unliganded dimers is not revealed to the readers although it would be appropriate, e.g., on top of page 5. Similarly, readers of the manuscript are not told about the fact in the main text that a Hill coefficient of 1 was used in some of the calculations. Although this fact is spelled out in the supplementary material, readers not conversant with EGF binding will not know that the Hill coefficient is not 1 for EGF binding to EGFR.

C. Original point #6: I raised a concern about f_{bound} being significantly larger than 1 for a large fraction of the vesicles in Fig. 5. According to the authors, this feature is due to white noise in the measurements. Anything that is not included in the fitted model will be interpreted as noise. My expectation is that the different expression level of EGFR in individual vesicles is a significant contributor to f_{bound} being larger than 1 since a vesicle with a high EGFR expression will bind a lot of EGF at high ligand concentration, more than vesicles with average EGFR expression. Since this has not been included in the calculations, it is interpreted as noise. My feeling is that correcting the bound EGF signal by the EGFR expression, i.e., TAMRA / mTurq, could significantly reduce the fraction of vesicles with f_{bound} larger than 1.

Reviewer #4 (Remarks to the Author):

I've now read the manuscript, looking specifically at the revisions and the responses from the authors to the comments of reviewer 2.

With regard to the comment 1: I agree with the reviewer that heterodimers as well as homodimers will most likely lead to a differential signaling response. The authors also recognise this and they correctly point out that the dimers will (in a heterozygous setting) be a mix of homodimers as well as heterodimers. Their results for the L858R mutation clearly show that the mutation in itself will have a big impact on the signaling response. In order to separate the signaling responses from homodimers, heterodimers can be the topic of a followup study, which seems fair given the inherent complexity of solely mapping the response of heterodimers and the lack of existing assays to do so.

With regard to comment 2: nomenclature is a topic heavily governed by personal preferences and will continue to be that way until an international standard has been established. Even within the GPCR field from which the article with reference 15 originated, there is an ongoing debate on the correct nomenclature and even with each new finding the nomenclature keeps changing. The authors have addressed the concern of reviewer 2 partially by removing intrinsic from the "mutation-induced" biased, but kept their preferred nomenclature for the "intrinsic ligand bias" as they named it - which seems reasonable.

For the final other/minor comments, the authors have expanded their discussion and 1) added sections

on system bias, 2) elaborated on ligand bias based on results in existing literature, and 3) explained the relative comparison of signalling responses of the system bias.

All in all, in my opinion the authors have made very significant progress in quantifying and comparing signalling responses of wildtype and mutated EGFR when bound to different stimuli as measured through site-specific phosphorylation. This makes it possible to better understand these responses and the impact of both natural as well as oncogenic mutations and stimulation by different (endogenous) ligand. Therefore, I would advise a reconsideration of the original decision.

RESPONSE TO REVIEWERS:

Reviewer #1 (Remarks to the Author):

....As this is in part a methodological advancement, I do think that more information regarding the use of the isolated osmotically produced vesicles needs to be added for clarity and accuracy of the methodology.

POINT 1. As I mentioned previously, and this was not addressed in this resubmission, the plasma membrane vesicles that are used in this study are not representative of the intact plasma membrane of cells, nor is there any data shown in this manuscript suggesting that they would be. The authors state: “Previous work has shown that the lipid composition is very similar to the lipid composition of the plasma membrane (19).” In the referenced work, the authors do not actually compare the lipid composition of these osmotically produced vesicles to the plasma membrane of cells, but rather to chemically produced plasma membrane vesicles. And even then, the referenced work highlights statistically significant changes between those two preparations. Specifically, the authors need to expand on their statement in the discussion: “These vesicles preserve many, although not all, characteristics of the plasma membrane.” The authors did add a statement discussing the loss of the cytoskeletal attachment and asymmetry of lipids, which is an improvement, but an elaboration of what is meant by which characteristics of these vesicles are similar to cellular plasma membranes should be also given.

I appreciate the authors’ comparison of these vesicles to nanodisks and suggest that what was written in the reviewer response be added to the discussion in the manuscript. I think the valid comparison here for this work is vesicles to nanodisks, not vesicles to intact, native plasma membranes of cells, and more emphasis throughout the work should be on this.

OUR RESPONSE. To address this concern, we have re-written a part of the Discussion, which now states:

“While these vesicles lack cytoskeleton and have perturbed asymmetry in their lipid composition, they allow access of macromolecules to both the extracellular and intracellular domain of the RTKs. In this respect, the plasma membrane derived vesicles can be considered as an alternative to nanodisks (33, 34). Unlike nanodisks, they do not impose artificial constraints on the free association of EGFR, and are a much more faithful mimic of the plasma membrane as they incorporate native lipids. They do not require RTK extraction out of the native plasma membrane and provide a contiguous membrane to ensure that the RTKs can associate with each other as they do in cells. Noteworthy, association constants measured for EGFR in vesicles and in cells are the same (26, 35). Also noteworthy, plasma membrane derived vesicles were recently leveraged in cryoEM studies to determine the high resolution structure of a membrane protein (8)”.

Reviewer #3 (Remarks to the Author):

POINT 1. When reading the first version of the manuscript, I raised issues about image segmentation (point 1) and the lack of correction for overspill and FRET (point 3). The authors convincingly rebutted this criticism in their response, and modified the manuscript accordingly. I also accept that bias itself can be seen from the bias plots that are generated from raw data. My other major concerns revolved around how realistic the calculations are given that they admittedly ignore significant aspects of EGFR dimerization. The fact that the calculations allowed the authors to reach the same conclusions as those based on bias plots does not verify the validity of the calculations themselves. While the authors attempt not to overinterpret them, still a significant fraction of the manuscript is devoted to these calculations. Specifically, limitation of the model used for correcting for unliganded dimers is not revealed to the readers although it would be appropriate, e.g., on top of page 5. Similarly, readers of the manuscript are not told about the fact in the main text that a Hill coefficient of 1 was used in some of the calculations. Although this fact is spelled out in the supplementary material, readers not conversant with EGF binding will not know that the Hill coefficient is not 1 for EGF binding to EGFR.

OUR RESPONSE. We agree, we omitted to mention in the Results that the Hill coefficient is set to 1, which is a requirement for the calculation of the bias coefficient β_{lig} . We have now added this fact in the Results, on page 5, and it was already discussed in the Supplement and in the Materials and Methods section of the main paper.

Point 2 . Original point #6: I raised a concern about f_{bound} being significantly larger than 1 for a large fraction of the vesicles in Fig. 5. According to the authors, this feature is due to white noise in the measurements. Anything that is not included in the fitted model will be interpreted as noise. My expectation is that the different expression level of EGFR in individual vesicles is a significant contributor to f_{bound} being larger than 1 since a vesicle with a high EGFR expression will bind a lot of EGF at high ligand concentration, more than vesicles with average EGFR expression. Since this has not been included in the calculations, it is interpreted as noise. My feeling is that correcting the bound EGF signal by the EGFR expression, i.e., TAMRA / mTurq, could significantly reduce the fraction of vesicles with f_{bound} larger than 1.

OUR RESPONSE. The reviewer is correct, the bound EGF signal must be divided by the EGFR expression signal. This is exactly how we have done it. We have added additional clarifications in the text. We have also clarified in the Figure 5 legend that both on the x axis and on the y axis we have ratios of measured fluorescence per unit membrane area.

Reviewer #4 had no suggestions, so no additional changes were made in the manuscript.

REVIEWERS' COMMENTS

Reviewer #1 (Remarks to the Author):

The authors have responded satisfactorily to my previous concerns.

Reviewer #3 (Remarks to the Author):

I agree with the response of the authors and happy to suggest acceptance of the manuscript.